# *Hand1* gene replacement with *Hand2* reveals overlap in function with unique occurrence of omphalocele and heart defects

Beth A. Firulli[1], Chloe A. Ferguson[1], Corrie de Gier-de Vries[2], Ram Podicheti[3], Douglas B. Rusch[3], Vincent M. Christoffels[2], Michael Rubart-von der Lohe[1] and Anthony B. Firulli[1,*]

## ABSTRACT

The bHLH transcription factors HAND1 and HAND2 are expressed in partially overlapping patterns during development. Studies have established evidence for significant functional redundancy between HAND1 and HAND2. To test redundancy fully, we engineered a *Hand1* allele in which we directly replaced the *Hand1* exons and intron with those of *Hand2*. The results show that 2% of *Hand1^{Hand2/Hand2}* mice are viable and fertile. The remaining *Hand1^{Hand2/Hand2}* embryos exhibit neonatal lethality due to omphalocele accompanied by ventricular septal defects and conduction anomalies. Omphalocele can occur due to altered gut rotation. Our transcriptomic expression analysis reveals that established gene expression patterns associated with normal gut rotation are compromised. Interrogation of cardiac function in surviving *Hand1^{Hand2/Hand2}* mice reveals QRS abnormalities and cardiac morphogenic defects. These data support previous findings that HAND factors exhibit extensive functional overlap but also reveals that HAND1 protein has unique functions within the *Hand1* expression domain and is required for normal embryonic development.

KEY WORDS: Gene replacement, HAND factor, Gut malrotation, Omphalocele, Transcriptomics, Heart development, Mouse

## INTRODUCTION

HAND1 and HAND2, which are basic helix-loop-helix (bHLH) transcription factors, have been established as crucial mediators of embryogenesis (George and Firulli, 2019; Massari and Murre, 2000; Murre, 2019; VanDusen and Firulli, 2012). HAND factors function by interacting with other bHLH proteins forming dimers that bind a canonical E-box (CANNTG) sequence that is well represented within the genome (Massari and Murre, 2000; Murre, 2019). HAND1 systemic knockouts die by embryonic day (E) 9.5 due to extra-embryonic defects within the visceral mesoderm of the yolk sac (Firulli et al., 1998; Riley et al., 1998). HAND2 knockouts die by E10.5 due to defects in pharyngeal mesoderm, which contributes to the second heart field (Srivastava et al., 1997; Tsuchihashi et al. 2011). Conditional knockouts employing various Cre drivers have demonstrated the importance of HAND factors in neural crest, endocardium, myocardium, epicardium and limb morphogenesis (VanDusen and Firulli, 2012; George and Firulli, 2019). Many of the encountered *Hand1* and *Hand2* conditional phenotypes occur within tissues in which both factors are co-expressed. The exceptions to this are the *Hand1* knockout phenotypes in the yolk sac and *Hand2* knockout phenotypes within the endocardium and epicardium where these factors are uniquely expressed (Barnes et al., 2011; Firulli et al., 1998; Riley et al., 1998; VanDusen et al., 2014b).

Studies interrogating the roles of HAND factors during development suggest that HAND factors are functionally redundant within co-expressed tissues (George and Firulli, 2019). Indeed, mutant phenotypes resulting from deletion of *Hand1* get more severe when *Hand2* gene dosage is reduced (Barbosa et al., 2007; Barnes et al., 2011; Vincentz et al., 2021). Similarly, *Hand2* left ventricle (lv) deletion phenotypes also become more severe when alleles of *Hand1* are reduced/removed (Vincentz et al., 2017). In gain-of-function analysis during limb development, both HAND1 and HAND2 cause polydactyly when overexpressed (Fernandez-Teran et al., 2003; McFadden et al., 2002). One can speculate from this data that the phenotypes of *Hand1* and *Hand2* loss-of-function and gain-of-function analyses result from overlapping function. However, recent studies of *Hand1* cardiomyocyte deletion models show survivable phenotypes that include ventricular septal defects (VSDs), malformed papillary muscles, and altered ventricular cardiac conduction morphology and function, suggesting unique roles for HAND1 in cardiomyocytes where HAND2 is co-expressed (Firulli et al., 2019; Vincentz et al., 2019).

To address the question of HAND factor functional redundancy rigorously, we engineered a replacement knock-in cassette that substitutes the *Hand1* coding domain and intron for that of *Hand2*. *Hand1^{Hand2/Hand2}* mice are viable; however, the frequency of *Hand1^{Hand2/Hand2}* pups is only 2%. *In utero* examination reveals normal Mendelian ratios, but 98% of *Hand1^{Hand2/Hand2}* embryos present with omphalocele and die neonatally. Surviving *Hand1^{Hand2/Hand2}* mice exhibit VSDs, small thin non-compacted ventricles, and abnormal cardiac conduction.

These data reveal that HAND1 and HAND2 are largely functionally redundant. The most notable evidence of this is the HAND2 rescue of the extra-embryonic defects caused by HAND1 loss of function; however, gut and heart phenotypes are encountered frequently, suggesting some unique protein functions. Supporting this, we observe that a single copy of *Hand2* is not sufficient for embryonic survival beyond E11.5. The discovery of omphalocele in the majority of *Hand1^{Hand2/Hand2}* neonates, combined with structural and conduction heart defects, suggests a unique role for HAND1 during gut and cardiac morphogenesis.

[1]Herman B Wells Center for Pediatric Research Department of Pediatrics, Anatomy, Biochemistry, and Medical and Molecular Genetics, Indiana University School of Medicine, 1044 W. Walnut Street, Indianapolis, IN 46202-5225, USA. [2]Department of Medical Biology, Academic Medical Center, University of Amsterdam, 22660, 1100 DD Amsterdam, The Netherlands. [3]Center for Genomics and Bioinformatics, Indiana University, Bloomington, IN 47405, USA.

*Author for correspondence (tfirulli@iu.edu)

A.B.F., 0000-0001-6687-8949

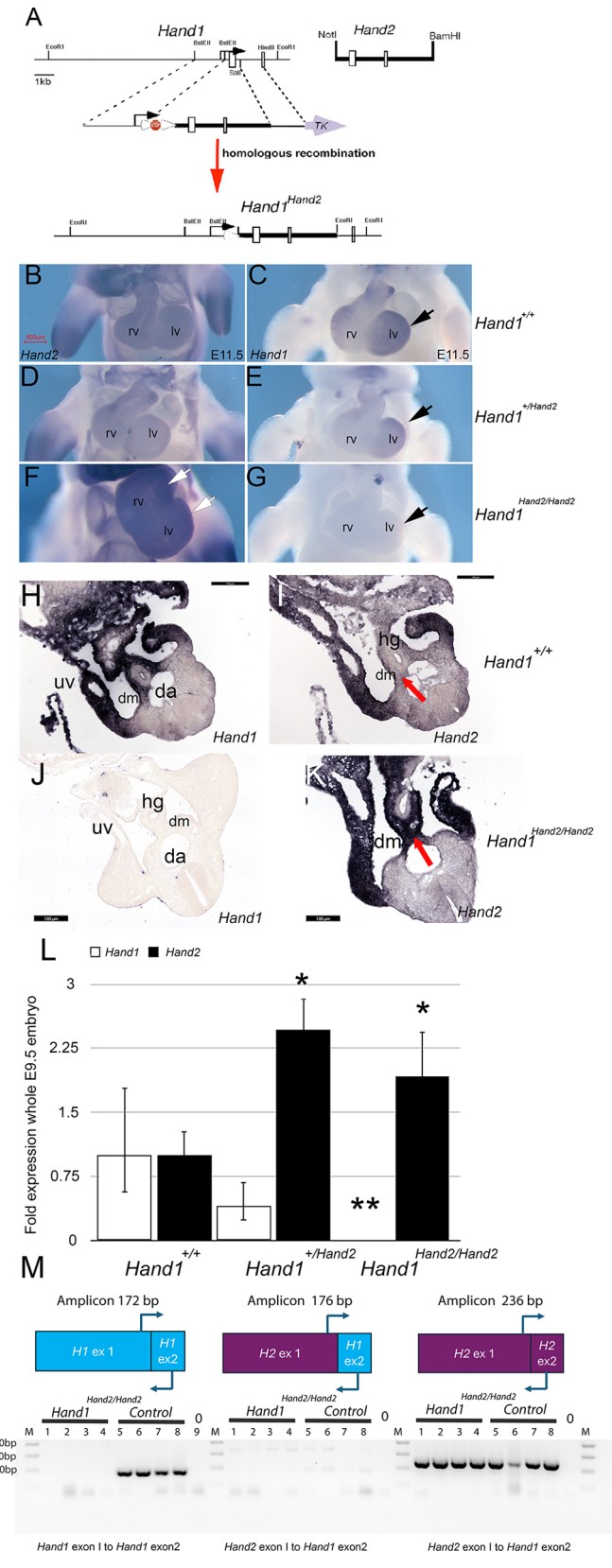

**Fig. 1. Gene targeting strategy and expression analysis of the Hand1^Hand2 replacement allele.** (A) *Hand2* 7.5 kb genomic sequence 3′ of its transcriptional start site and 5′ of its transitional start site including exon 1, intron, exon 2, and 3′ untranslated sequences flanked by established 5′ and 3′ *Hand1* targeting arms (Firulli et al., 1998, 2014, 2017b). The 3′ targeting arm of this allele contains the *Hand1* intron and 34 codons of *Hand1* exon 2. A stop-flox cassette was included for allele dormancy should it prove to be lethal. (B) E11.5 control (*Hand1^+/+*) *in situ* hybridization (ISH) showing *Hand2* cardiac expression. (C) E11.5 ISH showing *Hand1* cardiac expression that is more prominent in the left than the right ventricle (lv and rv, respectively). (D) ISH showing *Hand2* cardiac expression in *Hand1^+/Hand2* embryos. (E) ISH showing *Hand1* cardiac expression in *Hand1^+/Hand2* embryos. (F) ISH showing *Hand2* cardiac expression in *Hand1^Hand2/Hand2* embryos. (G) ISH showing *Hand1* cardiac expression in *Hand1^Hand2/Hand2* embryos. (H) E10.5 *Hand1^+/+* ISH for *Hand1*. Expression is robust within the lateral mesoderm dorsal mesentery (dm) and umbilical tissues surrounding the umbilical vein (uv). In C-G, arrows indicate *Hand1* lv expression. (I) E10.5 *Hand1^+/+* ISH for *Hand2*. Expression is robust within the lateral mesoderm but is lower within the dm and tissue surrounding the hindgut (hg). (J) E10.5 *Hand1^Hand2/Hand2* ISH for *Hand1* showing no *Hand1* expression. (K) E10.5 *Hand1^Hand2/Hand2* ISH for *Hand2*. *Hand2* transcripts are robustly detected in the dm and tissue surrounding the hg where *Hand1* expression is expected. (L) qRTPCR detecting *Hand1* (white bars) and *Hand2* (black bars) transcripts from E9.5 control (*Hand1^+/+*), *Hand1^+/Hand2* and *Hand1^Hand2/Hand2* whole embryos. Error bars represent the maximum and minimum expression observed within the cohort. n≥6. *P≤0.05, **P≤0.01 (Benjamini–Hochberg FDR). (M) RTPCR screen for 2-exon hybrid mRNA from E10.5 embryo/yolk sac cDNA using a sense primer from *Hand2* exon 1 and an antisense primer from *Hand1* exon 2. *Hand1* and *Hand2* amplicons confirm cDNA quality. da, dorsal aorta. Scale bars: 500 μm (B-G); 100 μm (H-K).

The construct was electroporated into mouse embryonic stem cells (ESCs) and germline transmission mice harboring the *Hand1^+/STOPHand2* allele were obtained. Male heterozygotes were crossed with *Tie2*-Cre females (de Lange et al., 2008) removing the stop-cassette and viable *Hand1^+/Hand2* mice were intercrossed to obtain *Hand1^Hand2/Hand2* embryos. No observable embryo phenotypes were detected at E10.5 within *Hand2^lacZ/Hand2* or *Hand1^Hand2/Hand2* embryos compared to E9.5 *Hand1^lacZ/lacZ* (Fig. S1A-C). *Hand1* and *Hand2* E11.5 wholemount *in situ* hybridization (ISH) of *Hand1^+/+* embryos showed modest levels of *Hand2* expression within both the right and left ventricles, and robust *Hand1* lv expression consistent with published studies (Fig. 1B,C) (Barnes et al., 2011; Firulli et al., 2010; Vincentz et al., 2017, 2021). Both *Hand1^+/Hand2* and *Hand1^Hand2/Hand2* E11.5 embryos showed an observable decrease in *Hand1* expression combined with an observable increase in *Hand2* within the lv (Fig. 1D-G). HAND gene expression within the caudal region of control E10.5 embryos was detected within the lateral mesoderm, the dorsal mesentery (dm), and surrounding the hind gut (Fig. 1H). *Hand2* expression was similar; however, expression within the dm and surrounding hind gut was noticeably lower than that of *Hand1* (Fig. 1I, red arrow). Examination of HAND gene expression in *Hand1^Hand2/Hand2* revealed undetectable *Hand1* expression whereas *Hand2* was observed to be robustly expressed within the dm and surrounding hind gut (Fig. 1J,K, red arrow). qRTPCR analysis from E9.5 whole embryos showed that detectable *Hand1* transcript was decreased within both *Hand1^+/Hand2* and *Hand1^Hand2/Hand2* embryos and detectable *Hand2* transcript was increased within *Hand1^+/Hand2* and *Hand1^Hand2/Hand2* embryos such that no *Hand1* message was detected within *Hand1^Hand2/Hand2* and *Hand2* expression was approximately twofold that of control expression (Fig. 1L). Based on construct design, the 3′ targeting arm required inclusion of *Hand1* exon 2 and, theoretically, a hybrid *Hand2* exon 1/*Hand1* exon 2 mRNA could be produced. This message would result in a frame-shift that could influence phenotype. We tested

## RESULTS

### Generation of a Hand2 replacement allele for the Hand1 locus

To explore HAND1 and HAND2 functional redundancy, we engineered a targeting construct that directly replaces the *Hand1* exons and intron with the exons and intron of *Hand2* (Fig. 1A).

for this hybrid message using E10.5 embryo/yolk sac cDNA and results showed no detectable hybrid message (Fig. 1M). A 3-exon hybrid mRNA is also possible; however, this mRNA codes for only HAND2 as translation terminates at the *Hand2* stop codon. We next interrogated RNA sequencing (RNA-seq) data (see below; Fig. 2) for both the 2- and 3-exon hybrid transcripts and found no reads of the 2-exon hybrid mRNA; however, the 3-exon hybrid message was detected in seven reads out of 125 million total RNA-seq reads, thus the replacement allele is functioning with minimal artifactual expression that could influence results.

### Transcriptomic analysis comparing *Hand1^Hand2/Hand2^* and H1CKO yolk sacs reveals rescue of the extra-embryonic *Hand1* loss-of function phenotype

*Hand1* systemic knockout embryos die by E9.5 from severe defects in the yolk sac visceral mesoderm and other extra-embryonic structures (Firulli et al., 1998; Riley et al., 1998) (Fig. S1A). Given that we observe phenotypically normal-looking E10.5 *Hand1^Hand2/Hand2^* embryos (Fig. S1C), we suspected that HAND2 was rescuing these HAND1 loss-of-function defects. To interrogate this, we performed RNA-seq analysis from E9.5 yolk sac RNA. We utilized the *Hand1* conditional loss-of-function allele (*Hand1^fx^*; McFadden et al., 2005) and the *Hand1* knockout allele (Firulli et al., 1998; McFadden et al., 2005) generating early mesodermal conditional *Hand1* knockouts (H1CKOs) by crossing *T-Cre; Hand1^lacZ/+^* (Perantoni et al., 2005) males to *Hand1^fx/fx^* females to produce *T-Cre;Hand1^fx/lacZ^* embryos (Fig. 2). Analysis was performed using RNA from controls (*Hand1^fx/+^*), H1CKOs (*T-Cre; Hand1^fx/lacZ^*), and *Hand1^Hand2/Hand2^* embryos. Results show that H1CKO yolk sacs exhibit a dynamic change in gene expression compared to controls, revealing numerous transcripts with false discovery rates (FDRs) of 0.05 (red) and 0.01 (blue) (Fig. 2A). In contrast, a volcano plot comparing *Hand1^Hand2/Hand2^* E9.5 yolk sac transcripts with control E9.5 yolk sac transcripts revealed a significantly subdued variation between the two transcriptomes.

Principal component analysis (PCA) showed robust variation between H1CKO yolk sacs (Fig. S2, red circles), control yolk sacs (*Hand1^fx/+^*; Fig. S2, green diamonds) and *Hand1^Hand2/Hand2^* gene replacement yolk sacs (Fig. S2, purple boxes). As expected, *Hand1* expression was downregulated in both *Hand1^Hand2/Hand2^* and H1CKO yolk sacs compared to controls (*H1^fx/+^*) (Fig. 2B). As indicated by the volcano plots, there were numerous genes misregulated in H1CKO yolk sacs and many of these exhibited a restoration of expression to control levels in *Hand1^Hand2/Hand2^* yolk sacs (complete RNA-seq differential expression data; Table S1). Examples of significantly regulated genes in H1CKO yolk sacs included *Smarca1*, which encodes a SWI/SNF family member that via its helicase and ATPase activities influences transcription by altering chromatin structure and plays important roles in cell reprogramming (Garry et al., 2021; Lickert et al., 2004; Peterson and Workman, 2000). *Smarca1* was significantly decreased in H1CKO yolk sacs ($-1.11$ LogFC, adjusted $P=1.23\times10^{-22}$), whereas *Smarca1* expression was notably restored within the *Hand1^Hand2/Hand2^* yolk sacs (Fig. 2B). Angiopoietin proteins such as angiopoietin 4 (Angpt4) play essential roles in vascular development and vessel formation (Ward and Dumont, 2002). RNA-seq analysis showed that *Angpt4* is significantly downregulated within H1CKO yolk sacs ($-2.4$ logFC, adjusted $P=1.35\times10^{-15}$; Fig. 2B). *Angpt4* expression was robustly restored and comparable to control levels within *Hand1^Hand2/Hand2^* yolk sacs (Fig. 2B). An example of a gene upregulated in H1CKO yolk sacs is placental growth factor (*Pgf*; 1.0 logFC, adjusted $P=0.019$; Fig. 2B). *Pgf* encodes a ligand for vascular endothelial growth factor receptor 1 (VEGFR-1; FLT1) and plays a role in modulating placental angiogenesis (Demira et al., 2007). Our results show that loss of *Hand1* results in an increase of *Pgf* expression that is rescued by replacing the missing *Hand1* alleles with *Hand2* alleles (Fig. 2B). Other interesting genes significantly regulated and rescued in *Hand1^Hand2/Hand2^* yolk sacs were: *Fgf8* (upregulated in H1CKO yolk sacs; 6.446 logFC, adjusted $P=7.71\times10^{-44}$; Fig. 2B); *Jag1* (upregulated in H1CKO yolk sacs; 2.533 logFC, adjusted

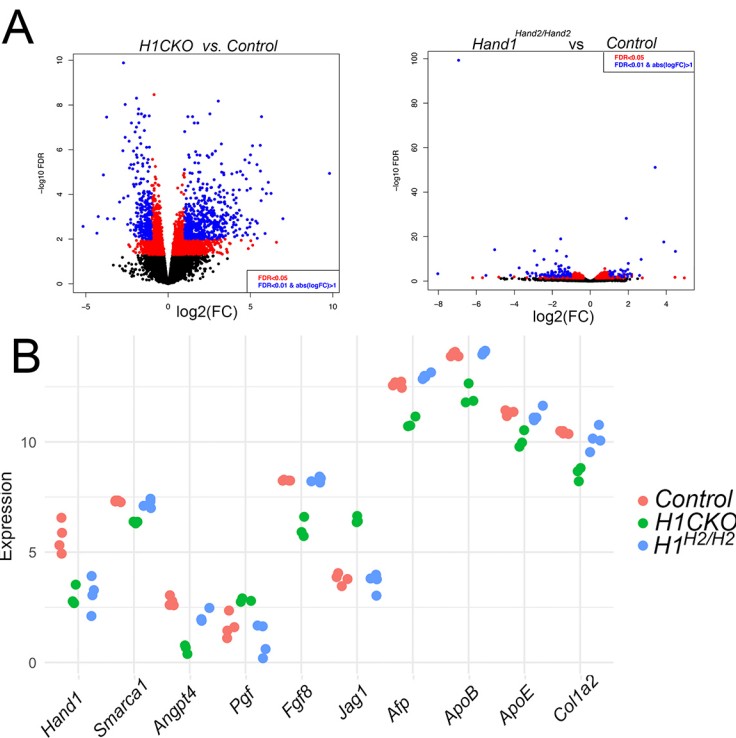

**Fig. 2. Gene expression in E9.5 T-Cre;*Hand1^fx/fx^* (H1CKO) yolk sacs are largely rescued in *Hand1^Hand2/Hand2^* embryos.** (A) Left: Volcano plot showing the comparison of gene expression between three H1CKO yolk sacs and four control (*Hand1^fx/+^*) yolk sacs. Black circles represent non-significant expression changes, red circles represent significant changes at a false discovery rate (FDR) of 0.05 and blue circles those at a significance FDR of 0.01. Right: Volcano plot showing the comparison of gene expression between four *Hand1^Hand2/Hand2^* and the four control yolk sacs. Gene expression comparison shows that the *Hand1^Hand2/Hand2^* transcriptome is similar to that of controls. (B) Individual expression analysis of ten genes within the three transcriptomes. *Hand1* expression is significantly downregulated in both H1CKO and *Hand1^Hand2/Hand2^* (*H1^H2/H2^*) yolk sac expression data. Each circle represents an individual yolk sac value. Controls, red; H1CKOs, green; *H1^H2/H2^*, blue.

$P=3.21\times10^{-33}$; Fig. 2B); α fetoprotein (downregulated in H1CKO yolk sacs; −1.89 logFC, adjusted $P=8.8\times10^{-31}$; Fig. 2B); apolipoproteins B and E (downregulated in H1CKO yolk sacs; −1.96 logFC, adjusted $P=3.39\times10^{-19}$; and −1.35 logFC, adjusted $P=1.29\times10^{-9}$ respectively; Fig. 2B) and *Col1a2* (downregulated in H1CKO yolk sacs; −1.99 logFC, adjusted $P=3.99\times10^{-15}$; Fig. 2B).

To observe changes in gene expression pathways between control and H1CKOs, we performed ingenuity pathway analysis (IPA; Table S2). Significantly altered IPA canonical pathways included 'calcium signaling' ($P=7.25\times10^{-10}$), as well as 'axonal guidance signaling' and 'synaptogenesis signaling pathway' ($P=3.75\times10^{-13}$ and $P=2.83\times10^{-10}$, respectively). Predicted upstream regulators included inhibition by HNF1A and HNF4A ($P=2.89\times10^{-18}$ and $P=2.83\times10^{-12}$, respectively) and activation by β-catenin (CTNNB1; $P=1.89\times10^{-15}$) and SOX2 ($P=4.07\times10^{-11}$). 'Cellular development', 'cellular morphology', 'cellular growth and proliferation', as well as 'cellular assembly and organization' were also significantly affected (Table S2). IPA results comparing controls and *Hand1^{Hand2/Hand2}* (Table S3) showed that the top canonical pathways distinctly affected in *Hand1^{Hand2/Hand2}* mutant yolk sacs are associated with cholesterol biosynthesis ('superpathway of cholesterol biosynthesis', $P=1.67\times10^{-12}$; 'cholesterol biosynthesis I, II and III', $P=2.72\times10^{-9}$) and 'superpathway of geranylgeranyldiphosphate biosynthesis I' ($P=1.77\times10^{-4}$), which plays a key role in post-translational modification of proteins. Predicted upstream regulators included the chaperone protein for cholesterol and lipids (SCAP; $P=4.9\times10^{-17}$), the lysophosphatidylcholine transporter MFSD2A ($P=2.58\times10^{-12}$) and the copper transporter gene ATP7B ($P=4.70\times10^{-11}$). Tissue development, organ morphology and organismal development systems were also altered in the *Hand1^{Hand2/Hand2}* yolk sacs. There was little overlap in the IPA categories altered in *Hand1^{Hand2/Hand2}* and H1CKO compared with controls, suggesting that HAND2 function compensates effectively for the loss of yolk sac HAND1. This was confirmed by the direct IPA comparison between H1CKO and *Hand1^{Hand2/Hand2}* yolk sacs (Table S4), which revealed similar significant alteration of IPA categories as those observed between control and H1CKO (Table S2). Together, these observations suggest that HAND2 yolk sac expression effectively replaces HAND1 function within the developing yolk sac, thereby rescuing HAND1 loss-of-function embryonic survival beyond E9.5.

### *Hand1^{Hand2/Hand2}* mice are viable at low penetrance

We next intercrossed *Hand1^{+/Hand2}* mice allowing pregnancies to go to birth. Genotyping of postnatal day (P) 10 neonates revealed a frequency of 0.02 for *Hand1^{Hand2/Hand2}* neonates (Table 1). The surviving 2% of *Hand1^{Hand2/Hand2}* neonates were viable and fertile (*n*=100); however, litter sizes from *Hand1^{Hand2/Hand2}* homozygous matings were only one to three pups on average. The significant decrease in *Hand1^{Hand2/Hand2}* frequency ($\chi^2=32.72$, $P<0.0001$; Table 1) suggests that the majority of *Hand1^{Hand2/Hand2}* mice die *in utero*, so we interrogated timed pregnancies to determine when the frequency of *Hand1^{Hand2/Hand2}* embryos decreased. Analysis of

*Hand1^{+/Hand2}* pregnancies from E10.5 to E18.5 revealed that *Hand1^{Hand2/Hand2}* embryos occur at a frequency of 0.25. We observed that in the majority of *Hand1^{Hand2/Hand2}* embryos at E16.5 the body wall surrounding the umbilicus was not fully closed and *Hand1^{Hand2/Hand2}* embryos/neonates presented with omphalocele (Fig. 3). Dissection of E16.5 embryos from a typical *Hand1^{+/Hand2}* intercross revealed intestines visible outside of the abdominal cavity of *Hand1^{Hand2/Hand2}* embryos (Fig. 3A). Histological sections through *Hand1^{+/Hand2}* control (Fig. 3B,D) and *Hand1^{Hand2/Hand2}* E16.5 littermates (Fig. 3C,E) confirmed that in the *Hand1^{Hand2/Hand2}* embryos both intestines and liver were observed outside the body wall at the level of the umbilicus (Fig. 3C,E, arrowheads). In most cases (Fig. 3C), there was no visible membrane surrounding the protruding organs. In some cases (Fig. 3E), a thin membrane surrounded the protruding organs, which is consistent with omphalocele.

Omphalocele is a congenital condition in which the muscle and connective tissues within the body wall surrounding the umbilicus are reduced resulting in a lack of umbilical ring closure that fails to contain the lower body organs (intestine and liver) and is

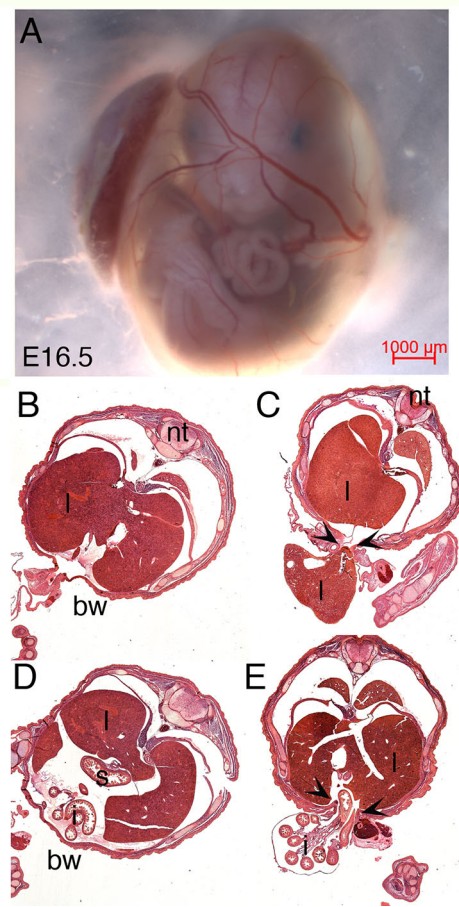

**Fig. 3. The majority of *Hand1^{Hand2/Hand2}* mice present with omphalocele.** (A) E16.5 wholemount *Hand1^{Hand2/Hand2}* embryo within the yolk sac. Intestines are visible outside the body wall. (B) Control *Hand1^{+/+}* H&E section showing normal body wall closure with the liver inside the body wall. (C) Matched section from a *Hand1^{Hand2/Hand2}* embryo revealing a widened opening at the umbilicus (arrowheads) with the liver outside the body wall. (D) Control *Hand1^{+/+}* H&E section showing normal body wall closure with the intestines inside the body wall. (E) Matched section from a *Hand1^{Hand2/Hand2}* embryo revealing a widened opening at the umbilicus (arrowheads) and intestines outside the body wall. bw, body wall; i, intestines; l, liver; nt, neural tube; s, stomach. Scale bar: 1000 µm.

**Table 1. Frequency of genotypes obtained from *Hand1^{+/Hand2}*×*Hand1^{+/Hand2}* intercrosses**

| Stage | *Hand1^{+/+}* | *Hand1^{+/Hand2}* | *Hand1^{Hand2/Hand2}* | *n* |
|---|---|---|---|---|
| E9.5 | 0.23 | 0.54 | 0.23 | 93 |
| E11.5 | 0.20 | 0.57 | 0.23 | 99 |
| E16.6 | 0.21 | 0.48 | 0.31 | 103 |
| P0-adult | 0.24 | 0.74 | 0.02* | 93 |

*$\chi^2$ 32.72, $P<0.0001$.

encountered in 1:4000 live births (Nichol et al., 2012; Poaty et al., 2019; Van Allen et al., 1987). To understand where the *Hand1* expression domain could be contributing to omphalocele in the *Hand1*$^{Hand2/Hand2}$ embryos beyond expression within the dorsal mesentery and gut (Fig. 1), we interrogated *Hand1* expression using our *Hand1*$^{lacZ/+}$ allele (Firulli et al., 1998) at E11.5 (Fig. S3). Frozen

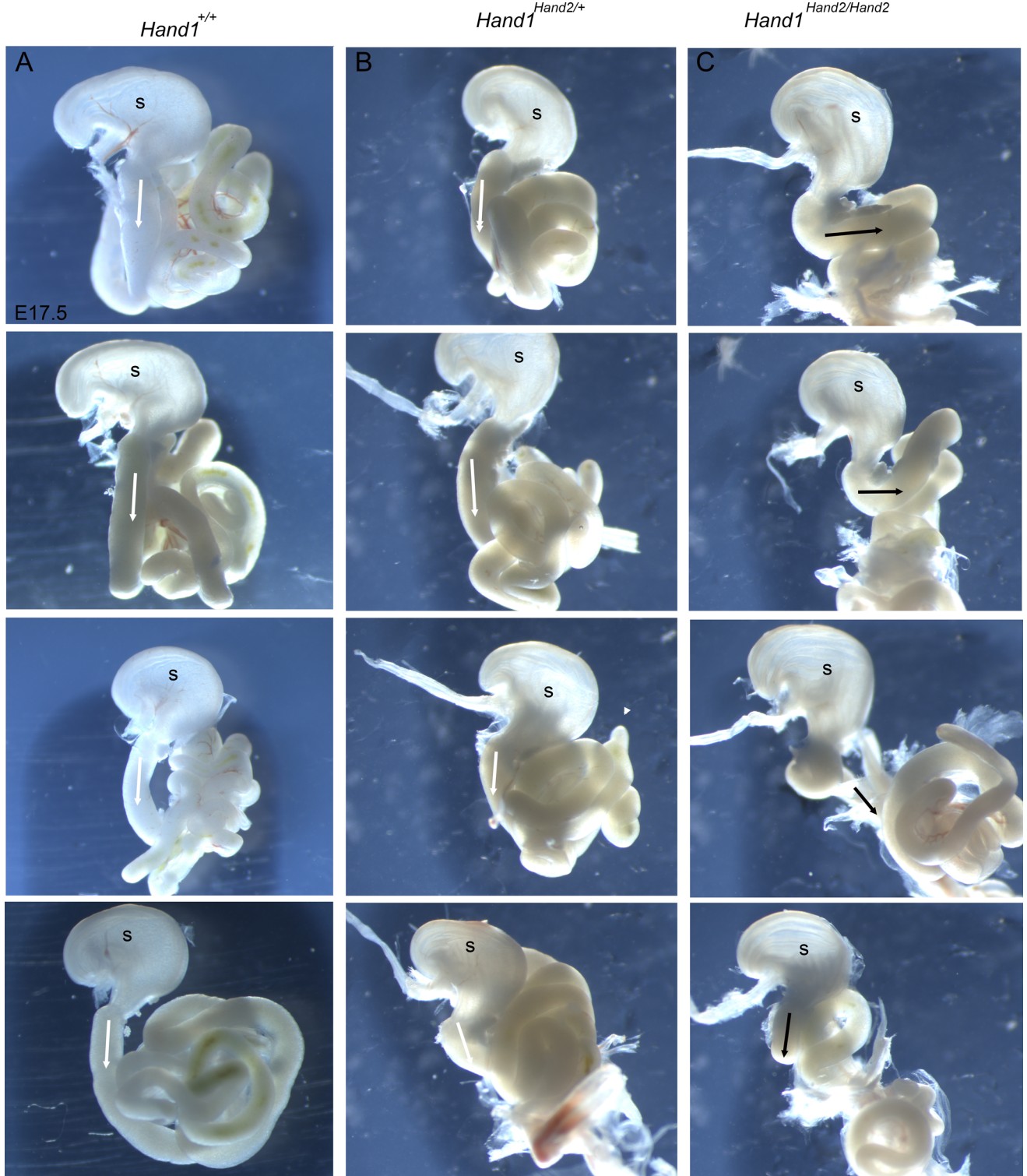

**Fig. 4. The majority of *Hand1*$^{Hand2/Hand2}$ mice present with a gut malrotation phenotype.** (A) Column of four control E17.5 gut preparations including the stomach (s) showing a consistent rotation whereby the intestine leaves the stomach in a downward direction (white arrows) before looping. (B) Column of four *Hand1*$^{Hand2/+}$ E17.5 embryo gut preparations revealing similar looping as controls (white arrows). (C) Column of four *Hand1*$^{Hand2/Hand2}$ E17.5 embryo gut preparations showing a deviation in gut looping whereby the intestines in two of the four examples make an extreme rightward turn at the level of the stomach (black arrows). A single example (third from top) exhibits a short downward section before the rightward loop, and the final example appears similar to controls. A total of 12 control, 14 *Hand1*$^{Hand2/+}$ and 14 *Hand1*$^{Hand2/Hand2}$ mutant embryos were observed from five litters.

transverse sections at the level of the umbilicus (arrows) revealed that there is robust X-gal staining within the ventral body wall in addition to expression within the smooth muscle of the intestines and within the umbilical tissue. At E12.5, X-gal staining within the body wall was diminished. To interrogate *Hand2* expression, we employed RNAscope on E13.5 sections to determine whether HAND gene expression is detectable beyond E12.5 and if *Hand2* is co-expressed (Fig. S4). The results showed that *Hand1* expression within the intestines marks smooth muscle and *Hand2* expression marks the enteric nervous system (D'Autreaux et al., 2011; Lei and Howard, 2011; Wu and Howard, 2002) (Fig. S4A,A'). *Hand1* and *Hand2* expression appeared to be non-overlapping (Fig. S4B-C', arrows) within the body wall surrounding the umbilicus.

Malrotation of the gut is an established mechanism causing omphalocele and gut rotation is in part mediated by left-right signaling involving the transcription factor *Pitx2* via *Wnt5a* signaling (Sanketi et al., 2022; Welsh et al., 2013). To determine whether there are defects in gut rotation, we fixed E17.5 embryos overnight and then carefully dissected away the body wall and tissues, finally removing the stomach and intestines for better imaging of the stomach and intestine looping morphology of control, *Hand1*^*Hand2/+*^ and *Hand1*^*Hand2/Hand2*^ embryos (Fig. 4). The results revealed that in four control and four *Hand1*^*Hand2/+*^ examples the gut tube extended straight down from the stomach before it began to loop (Fig. 4A,B, white arrows; total *n*=12 controls and *n*=14 heterozygotes). In contrast, *Hand1*^*Hand2/Hand2*^ looping was altered: a 90° rightward loop was observed in two of mutants,

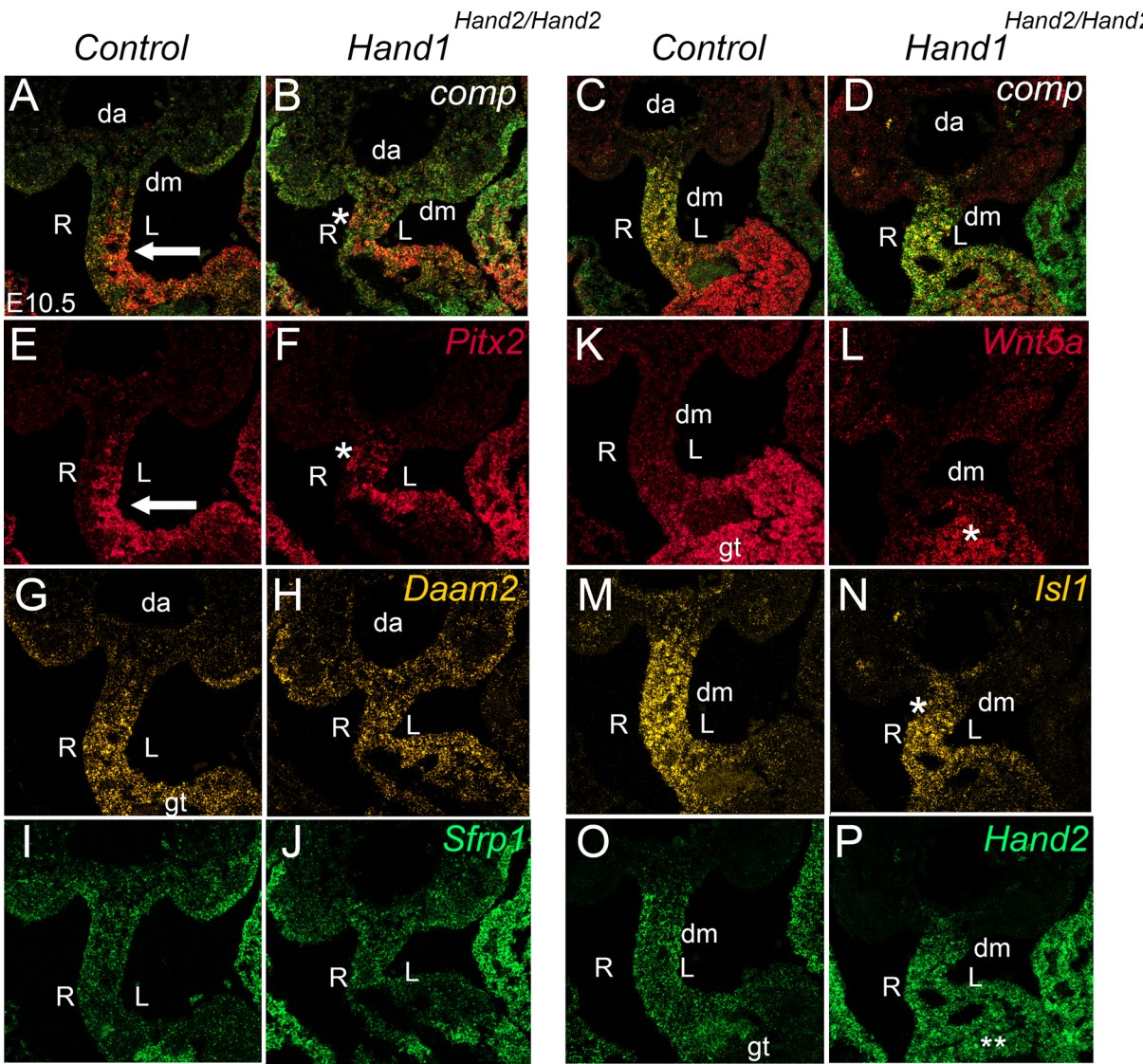

**Fig. 5. Altered gene expression within the dorsal mesentery of *Hand1*^*Hand2/Hand2*^ embryos.** (A,B) Composite expression of *Pitx2* (red), *Daam2* (yellow) and *Sfrp1* (green) in E10.5 control and *Hand1*^*Hand2/Hand2*^ embryos. (C,D) Composite expression of *Wnt5a* (red) *Isl1* (yellow) and *Hand2* (green) in control and *Hand1*^*Hand2/Hand2*^ embryos. (E,F) In controls, *Pitx2* expression is localized to the left (L) side of the dm, whereas in *Hand1*^*Hand2/Hand2*^ embryos *Pitx2* expression appears more uniform on both right (R) and left sides of the dm (asterisk). (G,H) Uniform expression of *Daam2* within both sides of the dm was observed in controls, but expression is more robust within the mesenchyme below the dorsal aorta within *Hand1*^*Hand2/Hand2*^ embryos. (I,J) *Sfrp1* expression within control and *Hand1*^*Hand2/Hand2*^ embryos. (K,L) Compared to controls, *Wnt5a* expression is markedly diminished (asterisk) within *Hand1*^*Hand2/Hand2*^ gut tube. (M,N) Compared to controls, *Isl1* expression is reduced (asterisk) in *Hand1*^*Hand2/Hand2*^ dm. (O,P) Compared to controls, *Hand2* expression is elevated in *Hand1*^*Hand2/Hand2*^ embryos within the gut tube (double asterisk) where *Hand1* is expressed (see Fig. 1). Three to five embryos were employed and evaluated per RNAscope probe. da, dorsal aorta; gt, gut tube.

one exhibited a milder rightward loop, and one appeared similar to controls (Fig. 4C, black arrows; total *n*=14 mutants). These data indicate that gut rotation appears to be altered within *Hand1^Hand2/Hand2^* embryos and contributes to the frequent omphalocele observed in the *Hand1^Hand2/Hand2^* embryos.

We next examined the expression of the left-sided marker *Pitx2* as well as *Daam2*, *Sfrp1*, *Wnt5a*, *Isl1* and *Hand2* expression at E10.5 when left-right pattern signaling for gut rotation is established (Sanketi et al., 2022; Welsh et al., 2013) (Fig. 5A-D). The results showed that in control embryos *Pitx2* expression is concentrated on the left side of the dm (Fig. 5E, arrow), whereas in *Hand1^Hand2/Hand2^* embryos *Pitx2* expression appeared more uniform throughout the dm (Fig. 5F, asterisk). Expression of *Daam2*, which encodes for a crucial mediator of WNT signaling, was distributed evenly through the dm and was more concentrated towards the gut tube of controls (Fig. 5G). *Daam2* expression in *Hand1^Hand2/Hand2^* embryos showed a similar pattern, but expression appeared to be upregulated within the mesoderm just ventral to the dorsal aorta (Fig. 5H). Secreted Frizzled related protein 1 (*Sfrp1*) also contributes to WNT signaling pathways. *Sfrp1* expression appeared similar in control and *Hand1^Hand2/Hand2^* embryos (Fig. 5I,J). Interestingly, expression of *Wnt5a* within the gut tube appeared to be markedly downregulated within *Hand1^Hand2/Hand2^* embryos compared to controls (Fig. 5K,L, asterisk). *Wnt5a* knockout mice present with gut malrotation defects (Welsh et al., 2013), suggesting the observed decrease in *Wnt5a* expression contributes to the observed omphalocele phenotype. The transcription factor ISL1 is also associated with regulation of gut rotation (Kurpios et al., 2008). Expression of *Isl1* was observed within the dm of both control and *Hand1^Hand2/Hand2^* embryos, but *Isl1* expression was visibly decreased within the *Hand1^Hand2/Hand2^* embryos (Fig. 5M,N, asterisk). *Hand2* expression appeared uniform within the dm and low within the gut tube tissues in control embryos, whereas *Hand2* gut tube expression in *Hand1^Hand2/Hand2^* embryos was visibly enriched in both the dm and gut tube where *Hand1* is robustly expressed (Fig. 5O,P, double asterisk; Fig. S1). Thus, several genes that encode crucial factors for gut looping are compromised in *Hand1^Hand2/Hand2^* embryos.

## Transcriptomic analysis of E13.5 embryos reveals that pathways associated with omphalocele are disrupted in *Hand1^Hand2/Hand2^* mutants

We initially employed a single-cell nuclei RNA-seq analysis from fixed tissue sections at the level of the umbilicus; however, analyses were inconclusive given the small number of *Hand1*-expressing cells relative to the non-*Hand1*-expressing cells within an E13.5 transverse section (Fig. S5). We therefore took a spatial transcriptomic approach from E13.5 control and *Hand1^Hand2/Hand2^* embryos whereby we captured the barcodes within the ventral-most tissue near the umbilicus (Fig. 6), focusing on regions within the ventral body wall and intestines based on *Hand1^lacZ/+^* expression (Fig. S3). All selected barcodes and images of Visium sections can be found in Tables S5 and S6. Results show two representative Visium slide sections marked for the selected barcodes captured for analysis (Fig. 6A). Uniform approximation and projection (UMAP) representation of the data collected from four *Hand1^Hand2/Hand2^* and four control embryos from multiple sections identified eight cell clusters (Fig. 6B, Table S7). Gene feature analysis revealed the distribution of *Hand2* and *Hand1* within each cluster and, as expected, expression of *Hand1* was significantly downregulated within the mutant clusters where *Hand1* is normally expressed (Fig. 6C, Table S7). Interestingly, *Hand2* expression was only slightly upregulated in Clusters 2 and 5 (<1 Log2FC, respectively) and

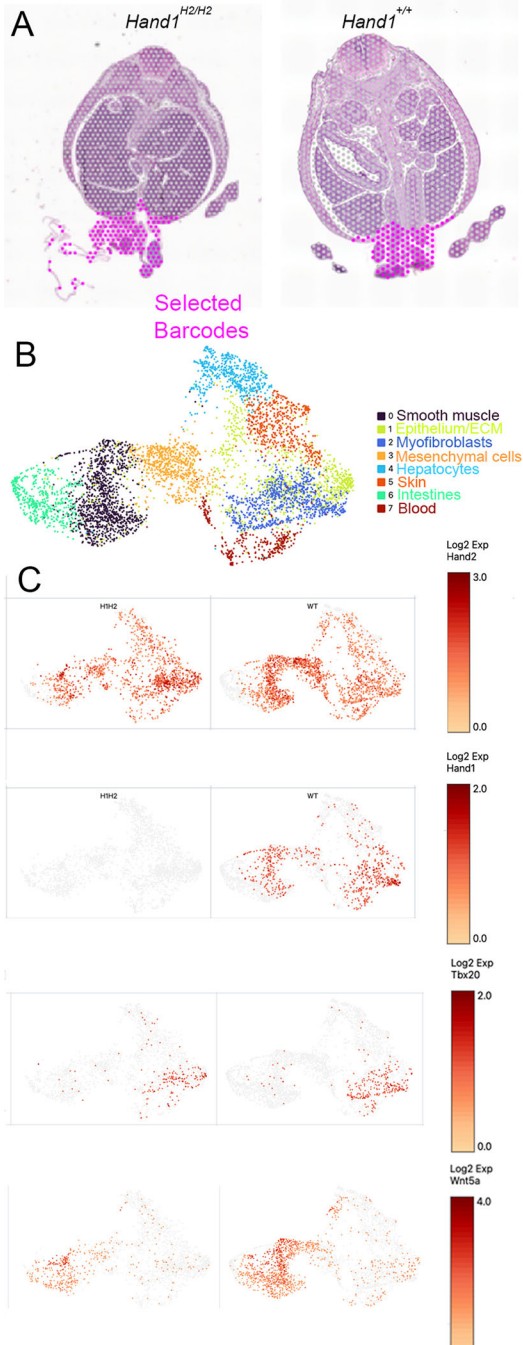

**Fig. 6. Spatial expression analysis of E13.5 of control and *Hand1^Hand2/Hand2^* embryos.** (A) Representative H&E images from Visium slides where barcodes were captured from the ventral body wall and tissues within and around the umbilicus from four *Hand1^Hand2/Hand2^* and four control E13.5 embryos (Tables S5 and S6). (B) UMAP representation of the captured bar codes with cluster identification established from marker gene expression in each cluster (see Table S7). (C) Gene feature expression images within the UMAP plot showing the key genes: *Hand2*, *Hand1*, *Tbx20* and *Wnt5a*. Detailed differential gene expression within each cluster can be found in Table S8.

was slightly down in Cluster 3 (−0.41 Log2FC, adjusted *P*=0.03). These data support that *Hand2* is not significantly overexpressed in the *Hand1^Hand2/Hand2^* mice. *Tbx20* expression exhibited strong overlap within myofibroblasts (Cluster 2); however, it did not show a significant decrease in the differentially expressed (DE) gene

analysis (Table S8). Supporting our E10.5 expression data, E13.5 expression of *Wnt5a* was reduced in *Hand1^{Hand2/Hand2}* compared to control embryos, being significantly downregulated in Cluster 1 (Log2FC− 2.03 adjusted $P$=1.51×10$^{-13}$; Fig. 6C, Table S8). This is significant as *Wnt5a* expression is essential for normal asymmetric gut morphogenesis (Sanketi et al., 2022; Welsh et al., 2013). Further DE gene analysis in Cluster 1 showed that *Pitx2* is significantly upregulated (Log2FC 1.05, adjusted $P$=2.1×10$^{-11}$) and *Ptch1*, a

component of the Shh pathway that is associated with omphalocele (Matsumaru et al., 2011), is significantly downregulated (Log2FC −1.86, adjusted $P$=9.45×10$^{-7}$; Table S8). Cluster 2 DE genes included an upregulation of *Hand2* (Log2FC 0.87, adjusted $P$=1.24×10$^{-15}$) as well a significant upregulation of *Pitx2* (Log2FC 0.49, adjusted $P$=2.88×10$^{-7}$). Cluster 3 also exhibited a significant upregulation of *Hand2* (Log2FC 0.73, adjusted $P$=0.03). Data from the remaining clusters are shown in Table S8. These data support

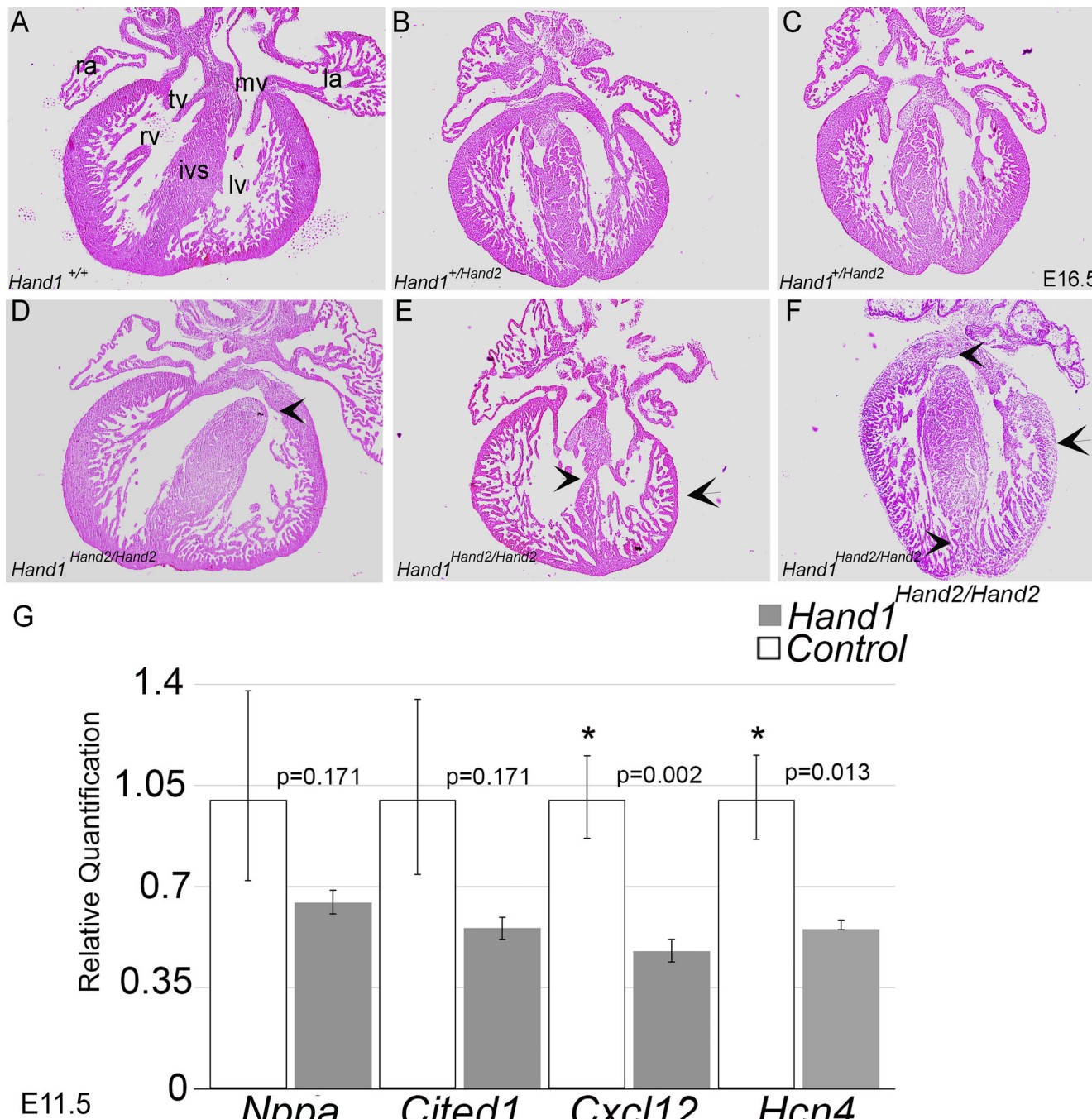

**Fig. 7. *Hand1^{Hand2/Hand2}* hearts display interventricular septum and free wall defects associated with altered gene expression.** (A) H&E section of an E16.5 control *Hand1^{+/+}* heart showing completed intraventricular septum (ivs) formation and normal compaction of both the right and left ventricle (rv and lv). Tricuspid (tv) and mitral valves (mv), right (ra) and left (la) atria are shown. (B,C) H&E matched sections from E16.5 *Hand2^{+/Hand2}* hearts. (D-F) Three matched sections from three individual E16.5 *Hand1^{Hand2/Hand2}* hearts. VSDs are visible in D and F (arrowheads) where a thin ivs and thin free walls indicative of poor compaction are visible in E (arrowheads). (G) qRTPCR from E11.5 control *Hand2^{+/+}* (white bars) and *Hand1^{Hand2/Hand2}* hearts (gray bars). Error bars represent the maximum and minimum relative quantification value obtained from each heart cohort (*n*≥6). *$P$≤0.05 (Benjamini–Hochberg FDR).

changes in pathways leading to gut malrotation, suggesting that altered left-right gene expression at E10.5 (Fig. 5) along with the continued decrease in *Wnt5a* expression and altered *Ptch1* expression occur in the absence of functional HAND1 protein.

### *Hand1^{Hand2/Hand2}* mutants exhibit cardiac malformations

Given that HAND factors play important roles in cardiac morphogenesis and cardiac conduction (Barnes et al., 2011; Firulli et al., 2019; VanDusen et al., 2014a; Vincentz et al., 2017, 2019, 2021), we examined E16.5 *Hand1^{Hand2/Hand2}* mutant hearts to determine whether cardiogenesis is compromised (Fig. 7). The results showed that control and *Hand1^{+/Hand2}* hearts appear normal with a fully developed outflow track, interventricular septum and compacted right and left ventricles (Fig. 7A-C). In contrast, *Hand1^{Hand2/Hand2}* E16.5 mutant hearts revealed VSDs, thin interventricular septum, and thin ventricular free wall with non-compacted myocardium (Fig. 7D-F). Heart size was not significantly different.

We previously published E11.5 *Hand1* cardiomyocyte deletion RNA-seq (Firulli et al., 2019). To look for possible HAND2 rescue of heart gene expression, we interrogated the expression of several genes significantly regulated in E11.5 *Hand1* cardiomyocyte deletion hearts (Firulli et al., 2019) by qRTPCR (Fig. 7G). Two examples of genes significantly regulated by HAND1 are atrial natriuretic factor (*Nppa*) (Lichardus et al., 1994) and *Cited1* (Bragança et al., 2019; McFadden et al., 2005). In *Hand1^{Hand2/Hand2}* hearts, although both genes showed a trend of downregulation, this did not reach statistical significance (Fig. 7G). In contrast, gene expression of a vascular maturation factor necessary for coronary development, *Cxcl12* (Cavallero et al., 2015; Ivins et al., 2015), as well as the first heart field and cardiac conduction system marker *Hcn4* (Liang et al., 2013), which are both significantly downregulated in H1CKO hearts (Firulli et al., 2019), remained significantly downregulated within *Hand1^{Hand2/Hand2}* mutant hearts (Fig. 7G). The observable cardiac phenotype combined with a partial rescue of HAND1-dependent gene expression suggests that HAND2 does not fully replace HAND1 function within the developing myocardium. Furthermore, the decreased levels of *Hcn4* suggest that cardiac conduction might be compromised.

### *Hand1^{Hand2/Hand2}* and *Hand1^{ΔLV/Hand2}* mice exhibit impairment of ventricular excitation

Deletion of the *Hand1* lv enhancer (*Hand1^{ΔLV/ΔLV}*) results in mice that exhibit both morphological and functional cardiac conduction defects (Vincentz et al., 2019). Intercross of surviving *Hand1^{Hand2/Hand2}* mice with a *Hand1^{lacZ/+}* (Firulli et al., 1998) allele resulted in no surviving *Hand1^{lacZ/Hand2}* neonates (Table 2). To determine whether this lethality is associated with extra-embryonic insufficiencies that a single *Hand1^{Hand2}* allele cannot rescue, we intercrossed *Hand1^{Hand2/Hand2}* mice with the *Hand1^{ΔLV}* allele (Vincentz et al., 2019). The results showed that all neonates (*n*=57) generated from this cross were the expected *Hand1^{ΔLV/Hand2}* genotype and were viable (Table 3). Importantly, no *Hand1^{ΔLV/Hand2}* mice presented with omphalocele.

**Table 2. Frequency of genotypes obtained from *Hand1^{Hand2/Hand2}* × *Hand1^{+/lacZ}* intercrosses**

| Stage | *Hand1^{+/Hand2}* | *Hand1^{Hand2/lacZ}* | n |
|---|---|---|---|
| E9.5 | 0.56 | 0.44 | 18 |
| E11.5 | ND | ND | ND |
| E16.6 | ND | ND | ND |
| P0-adult | 1.0 | 0 | 48 |

ND, not determined.

**Table 3. Frequency of genotypes obtained from *Hand1^{Hand2/Hand2}* × *Hand1^{ΔLV/ΔLV}* intercrosses**

| Stage | *Hand1^{ΔLV/Hand2}* | n |
|---|---|---|
| E9.5 | ND | ND |
| E11.5 | ND | ND |
| E16.6 | ND | ND |
| P0-adult | 1.0 | 57 |

ND, not determined.

To determine whether HAND2 replacement would rescue the observed *Hand1^{ΔLV/ΔLV}* conduction phenotypes (Vincentz et al., 2019), we interrogated cardiac conduction utilizing surface electrocardiogram (ECG) recordings on control (*R26R^{lacZ}*), *Hand1^{Hand2/Hand2}* and *Hand1^{ΔLV/Hand2}* mice (Fig. 8). All mice exhibited sinus rhythm during ECG recording. The median RR interval obtained from controls [146 ms, interquartile range (IQR)=136-154 ms] was not significantly different from that observed in *Hand1^{Hand2/Hand2}* mutant hearts (140 ms, IQR=126-157 ms; *P*=0.99 by Kruskal–Wallis one-way analysis of variance on ranks followed by Dunn's multiple comparisons; Fig. 8A); however, *Hand1^{ΔLV/Hand2}* mice exhibited a significantly shorter sinus node cycle length (122 ms, 117-131 ms) compared to controls (146 ms, IQR=136-154 ms; *P*=0.018 by Kruskal–Wallis one-way analysis of variance on ranks followed by Dunn's multiple comparisons) but the sinus cycle length of *Hand1^{ΔLV/Hand2}* mice was not significantly different from that observed in *Hand1^{Hand2/Hand2}* mice (140 ms, IQR=126-157 ms; *P*=0.063 by Kruskal–Wallis one-way analysis of variance on ranks followed by Dunn's multiple comparisons; Fig. 8A). This result indicates that the increased sinus node automaticity in *Hand1^{ΔLV/Hand2}* mice is due to alterations in the intrinsic sinus node properties and/or alterations in autonomic control of the sinus node.

*Hand1^{ΔLV/Hand2}* mice exhibited PQ(R) interval shortening in all three leads compared to both control and *Hand1^{Hand2/Hand2}* mice (Fig. 8B-D). For example, the mean PQ(R) interval of *Hand1^{ΔLV/Hand2}* mice in lead II (36±0.5 ms) was significantly shorter than that observed in both control and *Hand1^{Hand2/Hand2}* mice (42±1.7 and 40±0.5 ms, respectively; *P*<0.001 by ANOVA and Student–Newman–Keuls multiple comparisons). We did not observe significant differences when comparing control and *Hand1^{Hand2/Hand2}* mice (identical results were obtained for lead I and lead III; Fig. 8B-D). Overall, these results are indicative of accelerated atrio-ventricular conduction in *Hand1^{ΔLV/Hand2}* hearts.

The results for the QRS interval (Fig. 8E-J) were less straightforward to interpret. We observed significant differences in QRS1 and QRS2 intervals among the three genotypes, but not in all leads. Both median QRS1 and median QRS2 intervals in lead II were significantly longer in *Hand1^{Hand2/Hand2}* (QRS1: 9.9 ms, IQR=8.6-12.3 ms; QRS2: 11.9 ms; IQR=10.1-13.1 ms) and *Hand1^{ΔLV/Hand2}* mice (9.1 ms, IQR=8.6-9.9 ms; 11.1 ms, IQR=10.4-12.2 ms) compared to control mice (8.4 ms, IQR=7.7-8.9 ms; 10.0 ms, IQR=8.9-10.4 ms *P*<0.03 by Kruskal–Wallis test and Dunn's multiple comparisons; Fig. 8F,I); however, values were not significantly different between *Hand1^{Hand2/Hand2}* and *Hand1^{ΔLV/Hand2}* mice. The mean width of QRS1 in lead III (13.2±0.4 ms) of *Hand1^{Hand2/Hand2}* mice was significantly longer compared to both control and *Hand1^{ΔLV/Hand2}* mice (10.1±0.8 ms and 11.1±0.4 ms, respectively; *P*≤0.004 by ANOVA and Student–Newman–Keuls multiple comparisons; Fig. 8G). Overall, the electrocardiographic data suggests impairment of ventricular excitation in the *Hand1^{Hand2/Hand2}* and *Hand1^{ΔLV/Hand2}* hearts,

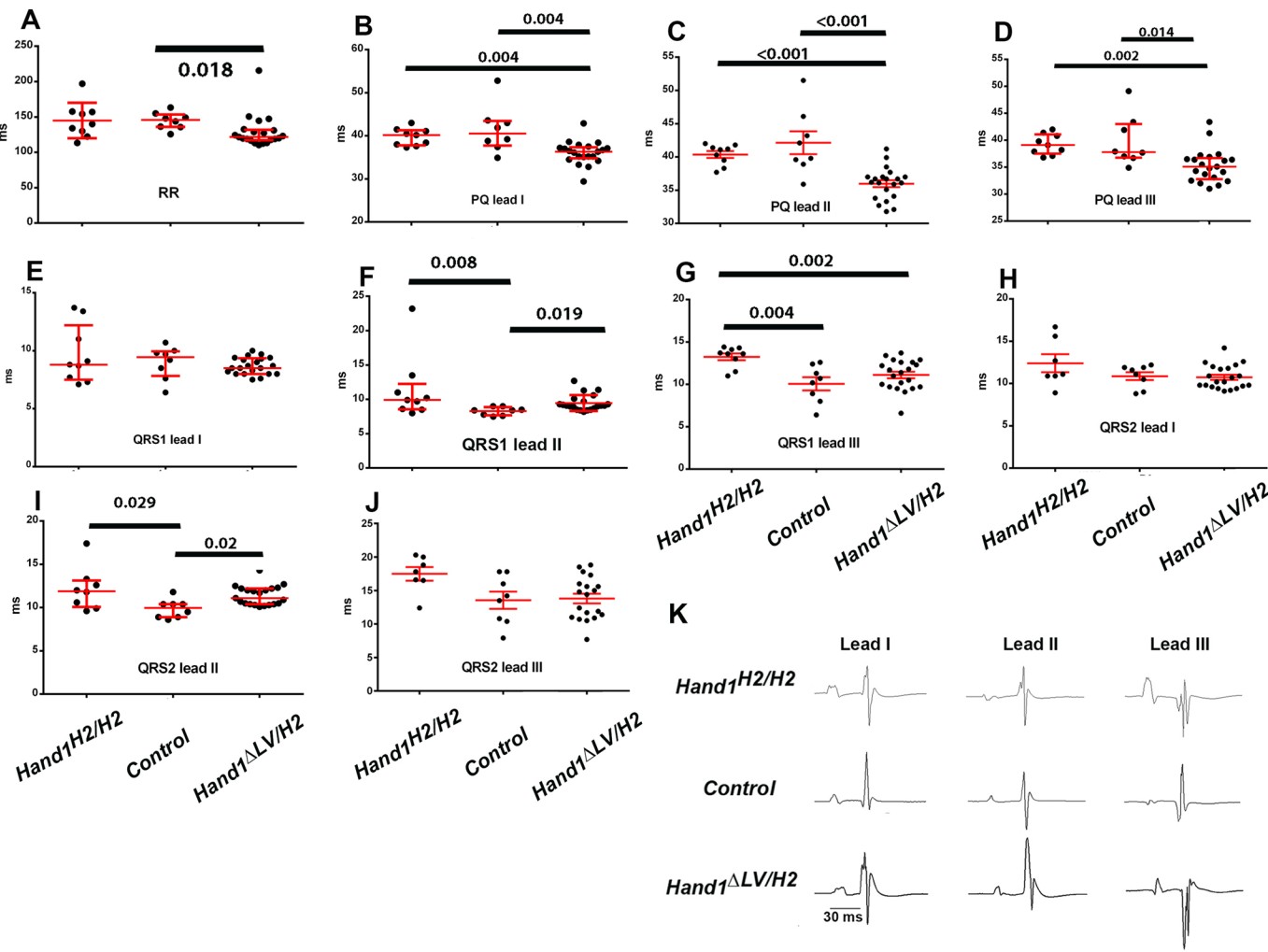

**Fig. 8. Electrophysiological phenotypes of control, *Hand1^{Hand2/Hand2}* and *Hand1^{ΔLV/Hand2}* mice.** (A-J) Dot plots of RR intervals (A), PQ intervals (B-D), QRS1 intervals (E-G) and QRS2 intervals (H-J). Error bars indicate median and interquartile range (A,B,D-F,I) or mean±s.e.m. (C,G,H,J). Numbers above black bars denote *P*-values by Kruskal–Wallis one-way analysis of variance on ranks followed by Dunn's multiple comparisons (A,B,D-F,I) or one-way ANOVA followed by Student–Newman–Keuls multiple comparisons (C,G,H,J). (K) Representative tracings from lead I, II, and III showing a single cardiac cycle from each genotype.

resulting from slowed propagation across structurally normal pathways and/or propagation across structurally altered conduction pathway, suggesting that HAND2 cannot fully rescue HAND1 cardiac conduction system (CCS) function. QT intervals revealed no significant changes between genotypes. Example surface ECG recordings from *Hand1^{Hand2/andH2}*, control and *Hand1^{ΔLV/Hand2}* are shown in Fig. 8K.

### *Hand1^{Hand2/Hand2}* conduction tissue appears morphologically normal

Given that the loss of *Hand1* lv expression results in visible morphological abnormalities due to His bundle, bundle branches and ventricular Purkinje networks (Vincentz et al., 2019), we sought to look for gross CCS morphology defects in *Hand1^{Hand2/Hand2}* mutant hearts using 3D-reconstruction analysis of *Hcn4* expression (Fig. 9). The results showed that conduction tissue appears similar between wild-type controls and *Hand1^{ΔLV/Hand2}* compound heterozygotes (Fig. 9A,B). Analysis of *Hand1^{Hand2/Hand2}* mutant hearts revealed that, other than morphological defects such as VSDs and thinner ventricle walls (Fig. 7), the morphology of the

conduction apparatus also did not appear to be obviously affected (Fig. 9C,D). Single images of each 3D reconstructed heart are shown below the reconstructed images noting the VSD (blue arrow) within the second *Hand1^{Hand2/Hand2}* mutant heart (Fig. 9E-H). Fig. S6 shows the sections employed in these reconstructions.

### DISCUSSION

In this study, we directly tested the functional redundancy of the transcription factors HAND1 and HAND2 using a replacement allele that substitutes HAND1 coding exons and its single intron with those of HAND2 (Fig. 1). HAND1 and HAND2 are highly similar proteins with a large overlap in their embryonic expression domains (George and Firulli, 2019; Vincentz et al., 2011). HAND factors bind similar *cis*-elements forming both hetero- and homodimers with themselves and other bHLH factors (Firulli et al., 2000, 2005, 2007). HAND factor gain-of-function studies show similar phenotypic outcomes within the developing limbs (Fernandez-Teran et al., 2003; McFadden et al., 2002) and loss-of-function phenotypes are most evident within tissues where *Hand1* or *Hand2* are uniquely expressed, such as the extra-embryonic

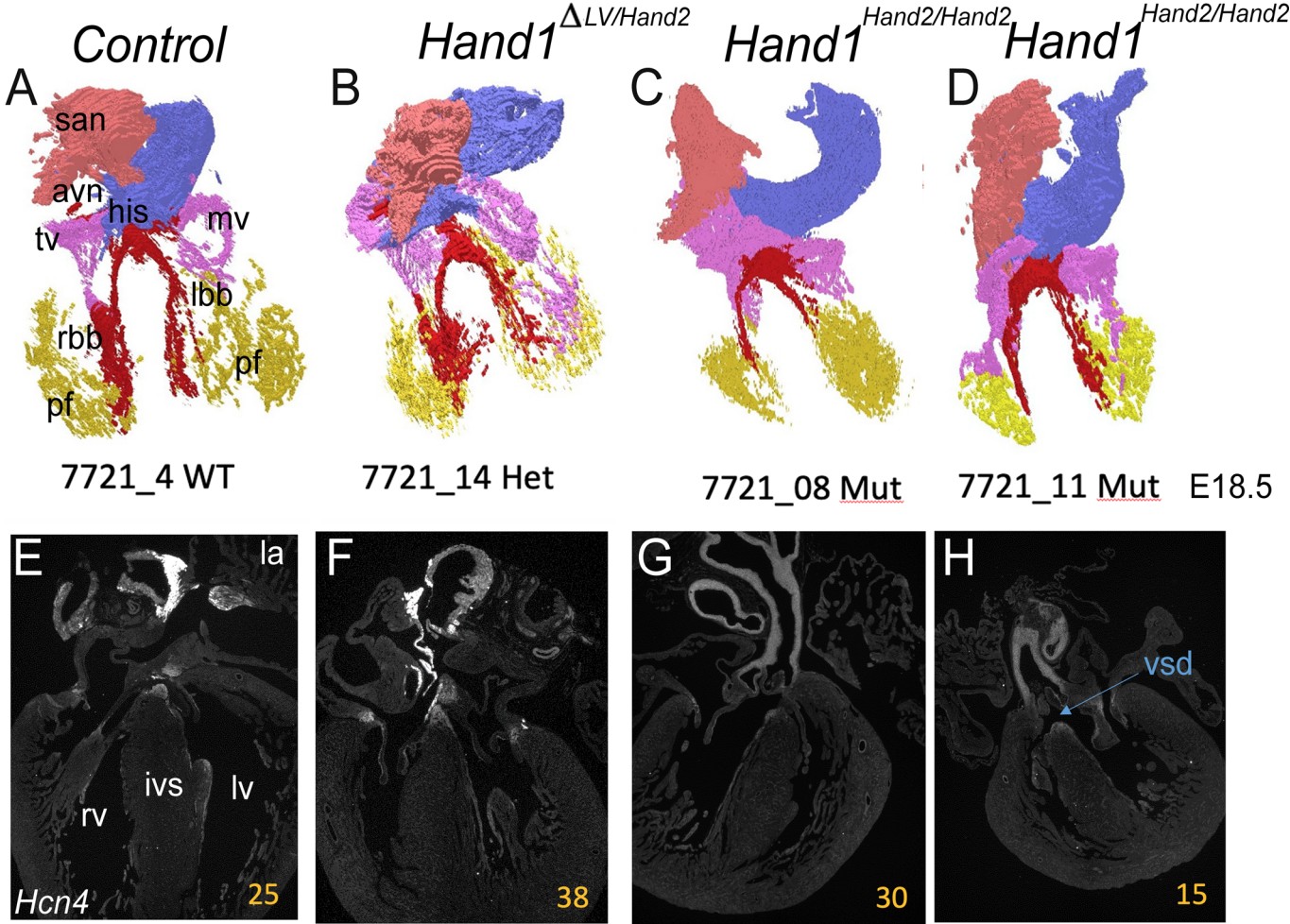

**Fig. 9. 3D rendering of the cardiac conduction system at E18.5.** (A) 3D reconstruction of a representative control heart conduction network. Sinoatrial node (san) is shown in salmon, atrioventricular node (avn) is shown in red and sits above the mitral valve (mv) and tricuspid valve (tv) shown in pink. His bundle (his), right (rbb) and left (lbb) bundle branches are shown in red outlining the interventricular septum (ivs) location. Purkinje fibers (pf) are shown in yellow. (B) 3D reconstruction of a representative *Hand1*^ΔLV/Hand2^ heart conduction network. No obvious structural abnormalities are observed. (C,D) 3D reconstruction of a representative *Hand1*^Hand2/Hand2^ heart conduction network. No obvious structural abnormalities are observed. Blue tissue is undefined. (E) Representative control section from *Hcn4* ISH that was used in the reconstruction of right ventricle (rv), ivs and lv showing normal cardiac morphology. (F) Representative *Hand1*^ΔLV/Hand2^ section from *Hcn4* ISH that was used in the reconstruction heart exhibits normal cardiac morphology. (G,H) Representative *Hand1*^Hand2/Hand2^ sections from *Hcn4* ISH that were used in the reconstruction of each heart. The heart in G shows normal morphology while the heart in H displays a VSD (blue arrow). Yellow numbers in E-H represent the section order for the 3D reconstructions.

mesoderm and CCS for *Hand1* (Firulli et al., 1998; Riley et al., 1998; Vincentz et al., 2019) and the second heart field mesoderm, epicardium, endocardium, and enteric nervous system for *Hand2* (Holler et al., 2010; Lei and Howard, 2011; Srivastava et al., 1997; Tsuchihashi et al., 2011; VanDusen et al., 2014a,b). Our results reveal that two copies of *Hand2* expressed from the *Hand1* locus robustly rescues the extra-embryonic defects observed within the systemic *Hand1* knockout. RNA-seq analysis reveals that most of the altered gene expression observed in *Hand1* mutants is restored; however, the majority of *Hand1*^Hand2/Hand2^ progeny die as neonates with omphalocele (Fig. 3, Table 1), suggesting that HAND2 cannot fully rescue HAND1 function within ventral body wall, dm and umbilical tissues. Mechanistically, omphalocele results from a malrotation of the gut during embryogenesis (Fig. 4) and/or body wall closure defects. Malrotation can be caused by alteration in left-right asymmetry pathways and indeed *Hand1*^Hand2/Hand2^ Visium and RNAscope expression analysis reveals changes in *Wnt5a* and *Pitx2* expression, which are known causes of omphalocele (Sanketi et al., 2022; Welsh et al., 2013) (Figs 5 and 6). Gene expression

within retinoic acid and BMP signaling pathways appear unaffected in the spatial data; however, the hedgehog receptor *Ptch1* is significantly downregulated within Cluster 1 (Table S8). This finding suggests that hedgehog signaling could contribute to the gut rotation phenotype as loss of SHH signaling has previously been identified as causative of omphalocele (Matsumaru et al., 2011; Negretti et al., 2022) and is a regulator of *Pitx2*; however, a direct SHH role in *Hand1*^Hand2/Hand2^ needs to be confirmed at earlier stages to make an informed conclusion.

Although 2% of *Hand1*^Hand2/Hand2^ mice are viable, 98% of *Hand1*^Hand2/Hand2^ mice exhibit omphalocele. Interestingly, *Hand1* conditional deletion employing *Tlx2-cre* also reveals an omphalocele phenotype (Maska et al., 2010). E9.5 *Tlx2-cre* expression is restricted to lateral mesoderm and in later stages marks the dm (Maska et al., 2010). Although both HAND factors are robustly expressed within E9.5 lateral mesoderm (Firulli et al., 1998; Riley et al., 1998; Srivastava et al., 1997) by E10.5 *Hand2* expression within the dm is low compared to *Hand1* expression (Fig. 1). We speculate that, within this tissue, HAND2 does not

sufficiently replace HAND1, resulting in the observed gut malrotation and subsequent omphalocele.

In the surviving $Hand1^{Hand2/Hand2}$ mice, the heart myocardium appears non-compacted accompanied with membranous and muscular VSDs (Fig. 7). $Hand2$ is expressed within the developing endocardium, ventricular myocardium, and epicardium (Barnes et al., 2011; George et al., 2023; VanDusen et al., 2014a; Vincentz et al., 2011) where $Hand1$ expression is only within the lv myocardium and myocardial cuff (Vincentz et al., 2019). Loss of myocardial or lv $Hand1$ expression is not lethal; however, $Nkx2.5^{Cre/+}$; $Hand1^{fx/fx}$ hearts exhibit VSDs, malformed papillary muscles and compaction defects (Firulli et al., 2019). $Hand1^{\Delta LV/\Delta LV}$ mice exhibit hypertrophic His bundle and Purkinje fibers accompanied by conduction defects (Vincentz et al., 2019). Loss of $Hand2$ within lv myocardium reveals no observable phenotypes; however, deleting both HAND genes within the lv results in cardiomyocyte occlusion of the lv lumen (Vincentz et al., 2017). Given that $Hand1$, $Hand2$ and $Hand1/Hand2$ loss-of-function phenotypes do not correlate well (beyond VSDs) with the observed $Hand1^{Hand2/Hand2}$ phenotypes, we suggest that myocardial HAND2 does not fully rescue loss of myocardial HAND1; the continued downregulation of $Cxcl12$ and $Hcn4$ within $Hand1^{Hand2/Hand2}$ hearts supports partial rescue. Based on previous association with HAND gene regulation, we speculate that HAND2 may be altering BMP/TGF, WNT and/or SHH signaling. Future studies of these pathways will be required to confirm either a gain-of-function or partial rescue mechanism.

ECG data confirm impairment of ventricular conduction both in the $Hand1^{\Delta LV/Hand2}$ and $Hand1^{Hand2/Hand2}$ hearts. We previously demonstrated that $Hand1^{\Delta LV/\Delta LV}$ mice exhibit QRS widening that was associated with deformities of the ventricular conduction system. In the current study, $Hand1^{Hand2/Hand2}$ and $Hand1^{\Delta LV/Hand2}$ mice (lead III) also exhibit QRS interval prolongations that were similar in magnitude to those of $Hand1^{\Delta LV/\Delta LV}$ mice (Vincentz et al., 2019), providing additional evidence that HAND2 cannot rescue the ventricular conduction phenotype associated with heart-specific loss of $Hand1$. In mice, changes in QRS morphology in the extremity leads are typically not predictive of specific ventricular conduction anomalies (e.g. right versus left bundle branch block). We are therefore unable to assign QRS changes seen in $Hand1^{Hand2/Hand2}$ and $Hand1^{\Delta LV/Hand2}$ mice to a specific conduction system defect. In contrast to $Hand1^{\Delta LV/\Delta LV}$ hearts, 3D reconstructions of the CCS do not reveal morphological conduction system defects within $Hand1^{Hand2/Hand2}$ or $Hand1^{\Delta LV/Hand2}$ hearts (Fig. 9), suggesting that functional alterations of the His-Purkinje network likely contribute to the conduction delay. Alternatively, structural and/or functional anomalies inhibiting impulse transmission at Purkinje-ventricular myocyte contacts may play a role. Finally, non-compaction of the ventricular myocardium is associated with ventricular conduction defects (Robida and Hajar, 1996), raising the possibility that the non-compaction seen in the $Hand1^{Hand2/Hand2}$ ventricles is causative of the QRS widening.

Based on construct design, two possible hybrid HAND mRNAs (H2ex1-H2ex2-H1ex2 or H2ex1-H1ex2) could contribute to phenotype. We searched for these hybrid mRNAs using RTPCR and RNA-seq evaluation and detected no 2-exon hybrid mRNAs. Bulk RNA-seq evaluation does detect rare 3-exon hybrid transcripts (seven reads out of 125 million reads). Three-exon hybrid transcripts will end translation at the $Hand2$ stop codon and thus only code for HAND2. It is therefore unlikely that hybrid HAND transcripts contribute significantly to the observed phenotypes. We also consider that body wall closure is an established omphalocele mechanism that is not mutually exclusive with gut malrotation. As

$Hand1$ expression within the ventral body wall is established and in zebrafish studies Hand2 plays an important mesothelial role (Prummel et al., 2022), these mechanisms must be considered in future studies.

In summary, HAND1 and HAND2 share a large functional overlap whereby HAND2 rescues the majority of $Hand1$ yolk sac defects but roles within the lateral and cardiac mesoderm-derived tissues are less conserved, revealing unique roles for HAND1 and/or revealing subtle differences in expression of the $Hand1^{Hand2}$ allele. Clearly, the role of HAND1 within the lateral mesoderm derivatives is important and the roles of HAND1 within asymmetric gene regulatory networks that modulate tissue looping outside of the heart may provide insights into congenital conditions such as omphalocele.

## MATERIALS AND METHODS

### Experimental mice

The $T$-$Cre$, (Perantoni et al., 2005) $Hand1^{lacZ}$, (Firulli et al., 1998) $Hand1^{fx}$, (McFadden et al., 2005) and $Hand1^{\Delta LV}$ (Vincentz et al., 2019) have been previously described and are summarized in Table S9. The $Hand1^{Hand2}$ allele was generated by the Indiana University Transgenic and Knock-Out Mouse Core. The $Hand1^{Hand2}$ allele was made using a targeting vector (Fig. 1) containing a 7.5 kb $Hand2$ genomic sequence beginning 3′ of its transcriptional start site and 5′ of its translational start site, including the $Hand2$ exon 1, intron, exon 2, and 3′ untranslated sequences. This cassette was flanked by established 5′ and 3′ $Hand1$ targeting arms (Firulli et al., 1998, 2014, 2017b). The 3′ targeting arm of this allele contains the $Hand1$ intron and 34 codons of $Hand1$ exon 2. Proper recombination results in replacing $Hand1$ exon 1 with $Hand2$ exon 1, intron and exon 2. A stop-flox cassette was included upstream of the $Hand2$ transcriptional start site. The 2.66 kb 5′ targeting arm and 1.4 kb 3′ targeting arm are nearly identical to those used in the systemic $Hand1$ knockout (Firulli et al., 1998). The targeting vector was electroporated into mouse ESCs, clones were selected, and targeting was determined by Southern blotting (12 kb wild-type EcoRI band; 4 kb $Hand1^{H2}$ band) as previously reported (Firulli et al., 1998, 2014, 2017a). PCR primers for the wild-type $Hand1$ allele (sense 5′-GGGAGGGACATAGGCGGGCGGGTTTT-3′ and antisense 5′-GGGGTCGGCGGGTGTGAGTGGTG-3′) were used as previously reported producing a 450 bp amplicon (Firulli et al., 2019). PCR primers specific for the $Hand1^{Hand2}$ allele (sense 5′-CGGAGGCCCTGTGCCTGGTGCTTCGTTTTGTG-3′ and antisense 5′-GGGCCCAGGGAAGACTCAAAACACC-3′) produce a 300 bp amplicon using the program cycle 94°C 2 min then 94°C 1 min, 60°C 1 min, 72°C 1 min for 35 cycles, followed by 72°C 10 min. The stop-flox cassette (Fig. S1) was initially added to ensure that $Hand1^{+/Hand2}$ mice would be viable should there be a gain-of-function phenotype from the inserted $Hand2$ allele. The cassette was removed within the germline using $Tie2$-$Cre$ females (de Lange et al., 2008). Heterozygous Hand1$^{+/Hand2}$ were encountered at Mendelian ratios, and all experiments were carried out with the systemically recombined $Hand1^{+/Hand2}$ allele that expresses $Hand2$ throughout development in regions where $Hand1$ is expressed. Genotyping of the $T$-$Cre$, $Hand1^{lacZ}$, $Hand1^{\Delta LV}$ and $Hand1^{fx}$ alleles were performed as described (Firulli et al., 1998; McFadden et al., 2005; Perantoni et al., 2005; Vincentz et al., 2019).

Animal stage choice rational employed in this study was as follows: E9.5 was chosen for yolk sac expression data (Fig. 2) as this is when $Hand1^{lacZ/lacZ}$ embryos begin to die. E16.5 was chosen for histological analysis for omphalocele (Fig. 3) as this is stage when body wall closure is completed. E17.5 (Fig. 4) was chosen to look at dissected stomach and intestine gut rotation as the process is completed by this stage. Left-right gene expression was observable at E10.5 within the dm and is an ideal time point for scoring early altered malrotation gene expression (Fig. 5). E13.5 is several days prior to body wall closure thus was the optimal time point for evaluating gene expression relevant to omphalocele (Fig. 6). E16.5 hearts were evaluated as intraventricular septum closure is fully completed by this stage (Fig. 7). Adult animals were employed for adult mouse conduction evaluation (Fig. 8). E18.5 hearts were used for 3D reconstructions (Fig. 9).

## Bulk RNA-seq library preparation and sequencing

Total RNA from four control ($Hand1^{fx/+}$), three H1CKO ($Hand1^{lacZ/fx}$) and four $Hand1^{Hand2/Hand2}$ ($H1^{H2/H2}$) yolks sacs was first evaluated for quantity and quality using an Agilent Bioanalyzer 2100. The RNA integrity number (RIN) across samples ranged from 3 to 6.4 with an average value of 4.69. Library preparation was performed using the KAPA mRNA Hyper Prep Kit according to the KAPA mRNA Hyper Prep Kit Technical Data Sheet, KR1352 – v4.17 (Roche). Sample libraries were then pooled in equal molarity, denatured, neutralized and applied to the cBot for flow cell deposition and cluster amplification. Sequencing was carried out with a HiSeq 4000 in 75 bp paired-end configuration (Illumina, Inc.). Approximately 30 million reads per library were generated. A Phred quality score (Q score) was used to measure the quality of sequencing. More than 94% of the sequencing reads reached Q30 (99.9% base call accuracy).

## Sequence alignment and differential expression analysis

The sequencing data were first assessed using FastQC (v.0.11.5, Babraham Bioinformatics) for quality control. All sequenced libraries were mapped to the mouse genome (UCSC mm10) using STAR RNA-seq aligner (v.2.5) (Dobin et al., 2013) with the following parameter: '–outSAMmapqUnique 60'. The read distribution across the genome was assessed using bamutils (from ngsutils v.0.5.9) (Breese and Liu, 2013). Uniquely mapped sequencing reads were assigned to mm10 refGene genes using featureCounts (subread v.1.5.1) (Liao et al., 2014) with the following parameters: '-s 2 –p –Q 10'. Each sample was analyzed independently and genes with read counts per million (CPM) <0.5 in more than the number of sample replicates in the smallest group were removed from the comparison. The data were normalized using the TMM (trimmed mean of M values) method. Multi-dimensional scaling analysis was performed with limma (v.3.38.3) (Ritchie et al., 2015). Differential expression analysis was performed using edgeR (v.3.24.3) (McCarthy et al., 2012; Robinson et al., 2010). FDR was computed from $P$-values using the Benjamini–Hochberg procedure.

## IPA enrichment analysis

Term enrichment, pathway and functional analysis were generated using IPA (QIAGEN Inc.; https://www.qiagenbioinformatics.com/products/ingenuity-pathway-analysis). Differentially expressed genes filtered for FDR≤0.01 and fold change absolute value ≥1.5 were passed to IPA. Genes were matched to the IPA knowledgebase and only matches specific to the reference species were used for analysis. Data are deposited in Gene Expression Omnibus (GEO) GSE223771.

## Single-cell RNA-seq and Visium spatial transcriptomics

Single-cell RNA-seq was performed using six 25 μm fixed sections from wax-embedded E13.5 control and $Hand1^{Hand2/Hand2}$ at the level of the umbilicus using the Chromium Fixed RNA Profiling (Gene Expression Flex) protocol from 10x Genomics following the company's detailed protocol. Sequenced reads from the Flex assay were demultiplexed, aligned to the mouse reference genome (mm10-2020-A) and the count matrices were generated using CellRanger v.8.0.1. Similarly, E13.5 control and $Hand1^{Hand2/Hand2}$ wax-embedded embryos were processed using Visium Spatial Gene Expression Reagent Kits for FFPE and sections were stained with Hematoxylin and Eosin (H&E) and imaged on a Keyence light microscope, then transferred onto Visium Spatial Gene Expression Slides using a Visium CytAssist (10x Genomics) following detailed protocols provided by 10x Genomics. Visium slides were then used to construct libraries that were sequenced by the Indiana University School of Medicine Center for Medical Genomics (CMG) core. Alignment of the sequenced reads from the Visium Spatial assay to the mouse reference genome (mm10-2020-A), tissue detection, barcode counting and feature identification were conducted using the spaceranger count pipeline v.3.0.1.

Further downstream analysis of the data generated by the Flex and the Visium Spatial assays were performed at the Center for Genomics and Bioinformatics at Indiana University, Bloomington, IN, USA. The barcode quality control was conducted through subjective assessment of the sample specific metrics: total UMI (unique molecular identifier) count per barcode, number of features per barcode and percentage of mitochondrial reads per

barcode generated by the package Seurat v.5.1.0 on R v.4.4.1. PCA was performed on the integrated sample data for dimensionality reduction, and the clusters were visualized using UMAP.

Results from Seurat analysis were exported to cloupe files using the R package loupeR v.1.1.1. Specific barcodes that contained ventral body wall and ventral gut structures were captured from four $Hand1^{Hand2/Hand2}$ and four $Hand1^{+/+}$ embryos (four to six sections per Visium slide; see Tables S5 and S6 showing barcodes and images of sections purple dots indicate selected regions). The identified barcodes were processed using Seurat v.5.1.0 using their recommended workflow in which the data were subjected to SCTransform-based normalization, sample merging, PCA and UMAP visualization to identify clusters. Eight distinct clusters of spatial barcodes were identified both containing wild-type and mutant cells. Significant DE genes within each cluster were called by applying the Wilcox test with FindMarkers function with a cutoff of adjusted $P≤0.05$.

Data from these analyses are deposited in GEO under accession numbers GSE299253 (for the Flex single-cell RNA-seq) and GSE298742 for the Visium data. Both data sets are also accessible in a super series entry: GSE299579.

## qRTPCR and RTPCR

qRTPCR was performed on a QuantStudio 3 (Applied Biosystems) quantitative thermocycler using TaqMan primers (Life Technologies) recognizing the following transcripts: $Hand1$, $Hand2$, $Cited1$, $Nppa$, $Hcn4$, $Cxcl12$. $Gapdh$ was used for normalization. Error bars denote the maximum and minimum relative level of gene expression in the test samples calculated using the confidence level set in the QuantStudio 3&5 software analysis settings. $P≤0.05$ generated by the QuantStudio 3 software, which calculates Benjamini–Hochberg FDR were regarded as significant and marked in all graphs with a single asterisk. Samples sizes of at least six were used in all experiments for all genotypes assayed. For Fig. 1M, $Hand2$-$Hand1$ hybrid message was not detected. $Hand2$ expression was detected in all samples. The 3-exon hybrid (H2 exon 1/H2 exon 2 H1/exon 2) codes for HAND2 as the $Hand2$ stop codon will terminate translation. Primer sequences used in Fig. 1M were: $Hand1$ exon 1 sense, 5′-CTGCGCCTGGCTACCAGTTAC-ATCGCC-3′; $Hand1$ exon 2 antisense, 5′-CCCCGAGGCAGGAGGGA-AGCTTTC-3′; $Hand2$ exon 1 sense, 5′-GACACTGCGCCTGGCCAC-CAGC-3′; $Hand2$ exon 2 antisense, 5′-TCACTGCTTGAGCTCCAGGGCC-CAG-3′. Cycles used: 94°C 2 min, then 94°C 1 min, 63°C 30 s, 72°C 30 s for 32 cycles. Final 72°C cycle 10 min. Amplicon size for $Hand1$ was 172 bp, 2-exon hybrid mRNA 176 bp, and $Hand2$ 226 bp.

## Histology

Embryos were fixed in 4% paraformaldehyde, dehydrated, embedded, sectioned, and stained with H&E as described (Firulli et al., 2010, 2014). A minimum of six viable embryos per genotype was used for all analyses. All data were collected on a Leica DM5000 B compound fluorescence microscope or Keyence BZ-X800 All-in-One fluorescence microscope.

## ISH and RNAscope

Wholemount and section ISH were performed as described (Firulli et al., 2017b; Vincentz et al., 2017). Antisense digoxygenin-labeled $Hand1$ and $Hand2$ riboprobes were synthesized using T7, T3 or SP6 polymerases (Promega) and DIG-Labeling Mix (Roche) using a linearized plasmid template as described (Firulli et al., 2017b; Vincentz et al., 2017).

RNAscope probes were obtained from Advanced Cell Diagnostics (ACD) and experiments were performed using the RNAscope® Multiplex Fluorescent Reagent Kit V2 (323100) along with the RNAscope® Probe –s. Target Probes for the following transcripts $Hand1$ (429651, C1, C2, C3), $Hand2$ (499821, C2, C3), $Pitx2$ (412841, C2, C3), $Daam2$ (1152921, C1), $Sfrp1$ (404981, C3), $Wnt5a$ (316791, C1) and $Isl1$ (451931, C2, C3) for simultaneous $in situ$ detection of two RNA targets in slide-mounted samples. All data were collected on a Zeiss LSM900 confocal microscope or Keyence light microscope.

## Immunohistochemical 3D reconstruction

3D reconstruction of HCN4 expression was performed as previously described (Soufan et al., 2003; Vincentz et al., 2019). Briefly, E18.5 hearts were

sectioned at 10 µm, and each third section was immunohistochemically stained with antibodies against both HCN4 and ACTC1. Images of immunohistochemically stained sections were then captured with a Leica DM 6000 fluorescence microscope. The stacks of images were aligned in Amira and then labeled in the same program to render a 3D reconstruction.

## ECGs

Surface ECGs were performed on mice that were lightly anesthetized with 2% isoflurane mixed with $O_2$. Mice were mounted on a heated stage with temperature and heartbeat monitored during recording. ECGs were recorded for 1 min at a sampling rate of 2000 Hz using the PowerLab26T (ADInstruments). $QT_c$ was calculated using LabChart software package (ADInstruments) using the equation $QT_c=QT/(RR/100)^{1/2}$, where QT is the measured Q-T interval and RR is the measured R-R interval (Mitchell et al., 1998). ECG intervals were measured by averaging 100 beats using LabChart software package. Statistical analysis was performed first by employing a Shapiro–Wilk test to test for normal distribution. If normal distribution was established, we next employed Student–Newman–Keuls multiple comparisons to determine significance. If the data were not normally distributed, the Kruskal–Wallis one-way analysis of variance on ranks was used followed by Dunn's multiple comparisons. Statistical significance is marked as: $*P \leq 0.05$, $**P \leq 0.01$.

## Acknowledgements
We thank Danny Carney, Kristy Thomson, Keelan Dyson and Harry Walker for excellent technical assistance. We thank Peter Nichols, Natasza Kurpios and Mark Kahn for scientific consultations on omphalocele and gut looping analysis. We thank the Center for Genomics and Bioinformatics, Indiana University, Bloomington, IN, USA, and the Center for Medical Genomics, Indianapolis, IN, USA. Infrastructural support at the Herman B Wells Center for Pediatric Research is in part supported by the generosity of the Riley Children's Foundation, Division of Pediatric Cardiology, and the Carrolton Buehl McCulloch Chair of Pediatrics.

## Competing interests
The authors declare no competing or financial interests.

## Author contributions
Conceptualization: B.A.F., A.B.F.; Data curation: B.A.F., C.A.F., C.d.G.-d.V., D.B.R., V.M.C., M.R.-v.d.L., A.B.F.; Formal analysis: C.A.F., R.P., D.B.R., V.M.C., M.R.-v.d.L., A.B.F.; Funding acquisition: A.B.F.; Investigation: B.A.F., C.A.F., A.B.F.; Methodology: B.A.F., C.A.F., C.d.G.-d.V., D.B.R., V.M.C., M.R.-v.d.L., A.B.F.; Project administration: A.B.F.; Software: R.P., D.B.R.; Supervision: B.A.F., V.M.C., A.B.F.; Validation: A.B.F.; Visualization: D.B.R.; Writing – original draft: A.B.F.; Writing – review & editing: B.A.F., D.B.R., A.B.F.

## Funding
This work is supported by the National Heart, Lung, and Blood Institute (2P01HL134599). Open Access funding provided by Indiana University-Purdue University Indianapolis. Deposited in PMC for immediate release.

## Data and resource availability
Sequence data have been deposited in GEO under accession numbers GSE223771, GSE299253, GSE298742 and GSE299579. All other relevant data and details of resources can be found within the article and its supplementary information.

## Peer review history
The peer review history is available online at https://journals.biologists.com/dev/lookup/doi/10.1242/dev.204963.reviewer-comments.pdf.

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
