## [Peer Review File · Development (Cambridge, England)]

***Hand1* gene replacement with *Hand2* reveals overlap in function with unique occurrence of omphalocele and heart defects**

Beth A. Firulli, Chloe A. Ferguson, Corrie de Gier-de Vries, Ram Podicheti, Douglas B. Rusch, Vincent M. Christoffels, Michael Rubart-von der Lohe and Anthony B. Firulli
DOI: 10.1242/dev.204963

Editor: Benoit Bruneau

Review timeline

Original submission:	20 May 2025
Editorial decision:	19 June 2025
First revision received:	28 August 2025
Editorial decision:	29 August 2025
Second revision received:	5 September 2025
Accepted:	8 September 2025

Original submission

First decision letter

MS ID#: dev.204963

MS TITLE: *Hand1* gene replacement with *Hand2* reveals overlap in function with unique occurrence of omphalocele and heart defects.

AUTHORS: Beth A Firulli; Chloe A Ferguson; Corrie de Gier-de Vries; Ram Podicheti; Douglas B Rusch; Vincent M Christoffels; Michael Rubart-von der Lohe; Anthony B Firulli

Dear Dr Firulli,

I have now received all the referees' reports on the above manuscript, and have reached a decision. The referees' comments are appended below, or you can access them online: please go to:

As you will see, the referees express considerable interest in your work, but have some significant criticisms and recommend a substantial revision of your manuscript before we can consider publication. If you are able to revise the manuscript along the lines suggested, which may involve further experiments, I will be happy receive a revised version of the manuscript. Your revised paper will be re-reviewed by one or more of the original referees, and acceptance of your manuscript will depend on your addressing satisfactorily the reviewers' major concerns. Please also note that Development will normally permit only one round of major revision. If it would be helpful, you are welcome to contact us to discuss your revision in greater detail. Please send us a point-by-point response indicating your plans for addressing the referees' comments, and we will look over this and provide further guidance.

Please attend to all of the reviewers' comments and ensure that you clearly highlight all changes made in the revised manuscript. Please avoid using 'Tracked changes' in Word files as these are lost in PDF conversion. I should be grateful if you would also provide a point-by-point response detailing how you have dealt with the points raised by the reviewers in the 'Response to Reviewers' box. If you do not agree with any of their criticisms or suggestions please explain clearly why this is so.

Reviewer 1

SUMMARY OF THE ADVANCE MADE IN THIS PAPER AND ITS POTENTIAL SIGNIFICANCE TO THE FIELD

In the manuscript by Firulli et al., the authors' goal is to test for functional redundancy of the Hand1/Hand2 paralogs. To do this, the authors generated a mouse line replacing the Hand1 gene with the Hand2 gene referred to as the Hand1^{Hand2} allele. Amazingly, many of the Hand1 knockout phenotypes are absent, including early yolk sac defects. Despite this, most homozygous Hand1^(Hand2/Hand2) animals die prior to or around birth with omphalocele, ventricular septal defects and conduction abnormalities. Molecular investigation into the omphalocele defect suggests that left-right patterning is disrupted alongside Hedgehog and non-canonical Wnt signaling. The few animals that survive to birth have electrophysiological abnormalities in the absence of morphological defects of the conduction system, although non-compaction phenotype of the ventricle exists. The authors conclude that the HAND1 and HAND2 proteins are partially, but not fully, redundant through this elegantly simple and straightforward genetic approach.

SUGGESTIONS TO AUTHORS

1. The animals are a little more complicated than the results section states. According to the methods, the coding portion of Hand1 exon 1 and most of the Hand1 intron were replaced with the Hand2 exons and intron. I can't tell from the diagram if the splice acceptor would still be present, but in any case, this leaves the second exon of Hand1 intact. Using the existing bulk transcriptional profiling data, is there any evidence that there is read through from the Hand2 transgene into the remainder of the Hand1 locus? Does the Hand2 exon1 from the transgene splice with exon 2 of Hand1? If so, what percentage of mature transcripts exhibit this splicing? How does this affect interpretation?
2. Loss of Hedgehog signaling early in development results in L/R patterning defects, including symmetrical expression of Pitx2. Given the loss of Ptch1/Wnt5a and gain of right-sided Pitx2, are the authors concluding that Hedgehog signaling is disrupted in gut development? From the spatial transcriptomics, is Shh/Ihh expression disrupted? Are there any factors up-stream of Hedgehog signaling (retinoic acid or BMP) disrupted in the spatial transcriptomics that may be direct targets of HAND1, but not HAND2, that may predict the root defect?
3. Given the authors extensive background in cardiac developmental defects of the Hand1/Hand2 mutants, can the authors expand their speculation in the discussion over genetic pathways likely abrogated by the Hand1-Hand2 switch that result in the cardiac non-compaction phenotype?

Reviewer 2

SUMMARY OF THE ADVANCE MADE IN THIS PAPER AND ITS POTENTIAL SIGNIFICANCE TO THE FIELD

The authors present significant and novel findings on these two important developmental transcription factors, HAND1 and HAND2. First, revealing important findings on both in vivo functional redundancies and also uncovering novel functions in intestinal development.

SUGGESTIONS TO AUTHORS

The manuscript by Firulli et al investigates the functional redundancy of the HAND1 and HAND2 transcription factors. Specifically, mice in which the Hand2 exons/intron were inserted into the Hand1 locus (replacing Hand1 exons/intron) were generated. Phenotypic and transcriptomic analysis demonstrated that Hand2 was able to rescue the extraembryonic defects found with complete deletion of Hand1 supporting redundancy in this aspect of Hand 1/2 functions. Interestingly, the subsequent analysis of these mice (Hand1Hand2/Hand2) demonstrated early lethality and an intestinal developmental abnormality, omphalocele, supporting a unique role for Hand1 in this developmental process. Molecular analysis using RNA in situ hybridization and spatial transcriptomics identified dysregulation of Wnt signaling and Pitx2 as potential mediators downstream of Hand1. Additional analysis of the hearts of E16.5 Hand1Hand2/Hand2 embryos demonstrated cardiac defects including thin myocardium and ventricular septal defects and mild cardiac conduction defects in adult survivors consistent with the inability of HAND2 to fully replace

HAND1 function in later stages of heart development and function. This was further supported by the identification of mild conduction abnormalities in mice harboring heterozygous for LV enhancer deletion of Hand1 and the Hand1Hand2 alleles (Hand1 Δ LV/Hand2). I have some minor comments which are detailed below.

1. Results section in manuscript and Table 1. Clarify results of intercrossing between Hand1+/Hand2 mice. Unclear why only 1-3 pups were found per litter as 25% should be expected to be Hand1Hand2/Hand2? Unclear if this is due to lethality of Hand1+/Hand2 mice or was the small litter size due to excessive maternal cannibalism?
2. Figure 3. Clarify the total number of embryos examined for gut rotation in the Figure legend and from how many litters these animals were collected.
3. Figure 4, re-organize the Control and Hand1Hand2/Hand2 images consistent for A-H and I-P so Control is always on left.
4. Figure 4, include the N for these experiments in the figure legend
6. Legend Figure 6A. Correct typo where "patent" interventricular septum is mentioned as septation is completed in control Hand1+/+ hearts.
7. Figure 6G. A select number of cardiac genes are examined at E11.5 by qRT-PCR, the addition of rationale for choosing these genes in manuscript text would be helpful for the reader
8. Table 3. Clarify the number of animals examined as in the Table n=50 but in the manuscript text n=57.
9. Fig. 7 and Supp Fig. 7. EKG abnormalities could be better represented to improve clarity. Some suggestions include: 1. Heart rate is described in manuscript text while associated Figure 7A shows on RR intervals, consider using one value to represent heart rate 2. Significance of QRS abnormalities that are specific to leads could be clarified, are they representative of right vs left bundle branch block and how do these findings compare to the phenotype of mice with loss of Hand1 in the conduction system.
10. Scale bars should be added for Supp. Figures 1B-1G, 2A-C,
11. Results, Section titled "Transcriptomic analysis comparing...". Define "Hand1fx" in text or Figure legend
12. Correct typo in Figure Legends. "Hand2Hand2/Hand2" should be "Hand1Hand2/Hand2"
13. Correct typo in manuscript, Results, section titled "Hand1Hand2/Hand2 mice are viable...", para 2, 2nd to last sentence. (ens; Figure 4A, A') should be Fig S5A, A')

Reviewer 3

SUMMARY OF THE ADVANCE MADE IN THIS PAPER AND ITS POTENTIAL SIGNIFICANCE TO THE FIELD

In their submitted manuscript, Firulli and colleagues present a Hand2 ORF substitution in the mouse Hand1 locus to address long-standing questions of redundancy between these two core transcription factors involved in cardiac and other development. The authors report several phenotype readouts under different genetic conditions. The data is complemented by spatial transcriptomics with main focus on omphalocele phenotypes connected to defects with ventral body wall closure.

SUGGESTIONS TO AUTHORS

In their submitted manuscript, Firulli and colleagues present a Hand2 ORF substitution in the mouse Hand1 locus to address long-standing questions of redundancy between these two core transcription factors involved in cardiac and other development. The authors report several phenotype readouts under different genetic conditions. The data is complemented by spatial transcriptomics with main focus on omphalocele phenotypes connected to defects with ventral body wall closure.

Overall, the manuscript adds to clarifying key unknowns in the field concerning the redundancy vs unique functions of Hand1 vs Hand2 during mammalian development. Nonetheless, in its current write-up, it unfortunately falls short of its intended purpose. More so, the manuscript is hard to follow as it jumps between different experiments, timepoints, allelic combinations, and tissue focus. For instance, the first paragraph of the intro has hardly anything to do with the overall scope of the work. Making better use of the intro space and discussion would already greatly improve the manuscript.

Big picture-wise, the authors state that their experiments show Hand1 and Hand2 to be broadly redundant - which is a claim that is not properly backed up by the presented data, as well as outlined with contradictory statements throughout the manuscript. For instance, the authors obtain only 2% survival with their substitution in a Hand1 null background: while the authors then go on to show this severely diminished outcome is due to intrauterine omphalocele and embryo loss, the context for their interpretation of redundancy is (as it seems) heavily influenced by the authors' deep knowledge of the topic and less by a rational chain of events outlined by the presented data. The mechanistic data on cardiac conductivity and function seem disconnected and are once more not placed in context to explain why these phenotypes are followed up.

Importantly, the paper write-up is highly specialized in its outline of used reagents, genetic backgrounds, readouts, as well as in how the resulting data is interpreted. With minimal to no introduction to previously established phenotypes associated with Hand1/Hand2 mutants, the manuscript is only accessible to deep experts. The figures are minimalistic at times and several critical data points are shoved into the supplemental data, where they do not belong (e.g., Supplementary Fig 1 is main figure material). Furthermore, several claims throughout the manuscript are interpretative and not presented with quantifications or comparisons to underline the significance. Many issues of the manuscript can potentially be salvaged, however, with a re-write and revisited presentation.

Additional Major Points:

- 1) Genetics: For their Hand1/2 substitution, the authors claim this is not an over-expression - but with an additional copy of Hand2 that is exactly what this is. The authors even explicitly state that there is a two-fold increase in Hand2 expression in the Hand1Hand2/Hand2 mice. This should be re-phrased and re-interpreted overall. This connects also to the overall statement as to how can these genes be functionally redundant when there's a 2% survival rate when you get rid of one of them?
- 2) Biological basics and background knowledge: Connecting to the challenging accessibility of the paper as presented, the figure showing the expression patterns of Hand1 and Hand2 in their respective mutants should be in the main paper and not a supplement. Furthermore, a proper outline of all used alleles and their phenotypes as originally reported, e.g., in the introduction (Hand1LacZ, etc.), would greatly strengthen the manuscript.
- 3) Phenotype interpretation: The authors claim the phenotype in the Hand1LacZ/LacZ is due to a yolk sac deficiency; however, they provide no evidence that this hypoplasia is solely due to the yolk sac deficiencies as opposed to any other developmental tissue.
- 4) Connection of the write-up with the data: throughout the paper, the text seems to be a train-of-thought text without incorporating all links to the actually obtained data. Case in point, the entire second paragraph of the section entitled "Transcriptomic analysis comparing Hand1Hand2/Hand2 and H1CKO yolk sacs reveals rescue of the extraembryonic Hand1 loss-of function phenotype.", the authors reference supp data but never once call out a figure or have that data represented in a main figure.
- 5) How meaningful are the interpretations of the intestinal curvature when dissected out of E17.5 mice? How faithful is the organ morphology when removed from the animal? Additionally, there seems to be an inconsistency among the control mice as well, such that interpreting differences in intestine looping feels dubious. In addition, the authors state this data matches with the observed high frequency of omphalocele in their mutants, but only observed a subtle curvature phenotype in 2 of 4 mice. Clarifying and contextualizing this data would greatly strengthen the work.
- 6) Related, the authors have previously contributed to establishing mesothelial expression of Hand1/2 - this tissue associates closely with ventral body wall closure (e.g., work from the Schultheiss lab in chick and mice), which the authors however seem to skip in favor of following anomalies with inner organs. Providing more context to the omphalocele phenotype would greatly expand the scope of the work and involvement of Hand1/2.

7) The gene expression panel following the spatial transcriptomics is presented without quantification of sample size or expression levels, but merely provided as overall interpretations. Quantification of expression levels (also compared to the Visium data) would strengthen the interpretation of these datapoints as well.

8) The cardiac function assays require better contextualization and connection with what has previously been shown about Hand function (which the authors know expertly well, but not every reader).

Minor points:

a) Developmental stages of performed experiments are widely variable, with several experiments being compared at different timepoints without explanation as to what the rationale is or how these time points fit together overall. Rationalizing the time points would greatly help in selling the manuscript to the reader.

b) The in-text description of Fig. 5B does not appear to be a complete sentence.

c) The method of gene expression evaluation is not mentioned (qPCR?). The authors are further encouraged to have a more specific label than "relative quantification" in 6G.

d) Figure 1 A, right label is overlapping with volcano plot (in downloaded high res version).

e) A few key typos of alleles, as well as inconsistencies in spelling of genetic features (lacZ vs LacZ, etc.).

First revision

Author response to reviewers' comments

Comments from the Reviewers:

Reviewer 1:

In the manuscript by Firulli et al., the authors' goal is to test for functional redundancy of the Hand1/Hand2 paralogs. To do this, the authors generated a mouse line replacing the Hand1 gene with the Hand2 gene referred to as the Hand1^{Hand2} allele. Amazingly, many of the Hand1 knockout phenotypes are absent, including early yolk sac defects. Despite this, most homozygous Hand1^(Hand2/Hand2) animals die prior to or around birth with omphalocele, ventricular septal defects and conduction abnormalities. Molecular investigation into the omphalocele defect suggests that left-right patterning is disrupted alongside Hedgehog and non-canonical Wnt signaling. The few animals that survive to birth have electrophysiological abnormalities in the absence of morphological defects of the conduction system, although non-compaction phenotype of the ventricle exists. The authors conclude that the HAND1 and HAND2 proteins are partially, but not fully, redundant through this elegantly simple and straightforward genetic approach.

We thank Reviewer 1 for these highly supportive and encouraging comments.

1. *The animals are a little more complicated than the results section states. According to the methods, the coding portion of Hand1 exon 1 and most of the Hand1 intron were replaced with the Hand2 exons and intron. I can't tell from the diagram if the splice acceptor would still be present, but in any case, this leaves the second exon of Hand1 intact.*

Using the existing bulk transcriptional profiling data, is there any evidence that there is read through from the Hand2 transgene into the remainder of the Hand1 locus? Does the Hand2

exon1 from the transgene splice with exon 2 of Hand1? If so, what percentage of mature transcripts exhibit this splicing? How does this affect interpretation?

We thank Reviewer 1 for pointing this very significant issue out. Indeed, we had to employ the *Hand1* intron and a portion of *Hand1*, exon II as a 3' targeting arm. The cassette contains a significant amount of *Hand2* 3' UTR in the cassette including the *Hand2* polyadenylation sequences. Given we did not rule out that some percent of transcripts could be hybrids that could contain *Hand1* exon 2 (*H1EX2*) either a 3-exon transcript where *Hand2* exon 1 (*H2EX1*) splices to *Hand2* exon 2 (*H2EX2*) and then to *H1EX2*, thus *H2EX1-H2EX2-H1EX2*. This 3-exon message would only produce HAND2 protein given its termination codon would stop translation of the frame and there is not ATG within the HAND1 exon 2 codons.

Figure showing RT-PCR from E10.5 embryos and yolk sacs for wild type *Hand1*, the *Hand1-Hand2* hybrid, and wild type *Hand2* message.

The *H2EX1-H1EX2* would be more troubling as it would result in a frame shift (see below). To test the possibility of the two exon hybrid *H2EX1-H1EX2*, we designed an RT PCR analysis using sense primers from *Hand2* Exon I, and an antisense *Hand1* exon 2 primer to look for amplicons that would indicate that this possible message is processed. In this experiment, we employ cDNA from E10.5 embryo and yolk sacs. Results show that we do not detect *H2EX1-H1EX2* hybrid mRNA (Figure above). Our informaticians searched for the *H2EX1-HEX21* hybrid message and this message was not detected. However, the 3 exon mRNA (*Hand2* exons 1 & 2 spliced to *Hand1* exon 2) was detected at 7 reads out of 125 million reads. As stated above *HAND2* would be normally translated by the transcript as the *Hand2* stop-codon would stop translation. We hope the direct assay from cDNA that we present above combined with the suggested screen of the bulk RNA-seq data shows that we do not detect the troublesome 2-Exon Hybrid but do observe a very low level of the 3-exon hybrid message which as stated will only code for *HAND2*. In the revised manuscript, we now define the nature of our construct with more clarity, add the included RT-PCR within Figure S1, discuss that we do observe the 3-exon message at a low frequency within the bulk RNA-Seq data and hope these changes satisfactorily address this important concern and thank the Reviewer again for thinking of this.

For the Reviewer below is the hypothetical *H2EX1-H1EX2* frameshift transcript. *Hand2* ex 1 (green)- *Hand1* ex 2 (red) frame-shift. The codon shift is shown in black. Given we do not detect this transcript we feel adding this to the supplemental data would be too confusing to the reader.

Hand2 exon 1. hand1 exon real frame **BOLD** Frameshift below in black
 agg aag aaa gag ctC CTC AGC AGC Ccg aaa get tcc ctc ctg cct
 a R K K E L P Q Q P E S F P P A
 L S S P K A S L L P
 cgg ggc ccg gcg aga aga gga tta aag ggc gca ccg gct ggc ctc agc
S G P. G E K R I K G R T G W P
 R G P A R R G L K G A P A G L
 H1 stop out of frame
 aag tct ggg cgc tgg agc taa acc agt gag
Q Q V W A L E L N Q *
 K S G R Y S ***

2. *Loss of Hedgehog signaling early in development results in L/R patterning defects, including symmetrical expression of Pitx2. Given the loss of Ptch1/Wnt5a and gain of right-sided Pitx2, are the authors concluding that Hedgehog signaling is disrupted in gut development? From the spatial transcriptomics, is Shh/Ihh expression disrupted? Are there any factors up-stream of Hedgehog signaling (retinoic acid or BMP) disrupted in the spatial transcriptomics that may be direct targets of HAND1, but not HAND2, that may predict the root defect?*

This is also great question. From the DE analysis performed on the Visium spatial transcriptomics *Shh*, and *Ihh* are not significantly altered in expression between mutants and controls nor is any *Bmp*, *Rara*, *Rarb* or *Rxra* (Firulli 2025 Supplemental file 8). Given we observe only *Ptch1* altered, we were reluctant to strongly suggest HH signaling could be significantly altered and we now speculate some within the discussion focusing on downregulation of *Ptch1* as requiring follow-up. We do recognize that we only have an E13.5 Visium datasets to examine this, we cannot rule out that changes in hedgehog, RA, and BMP gene expression within earlier time points of the gut and body wall, points. We feel this is certainly something to follow up on in our next manuscript and hope our edits satisfy the Reviewer's concerns.

3. *Given the authors extensive background in cardiac developmental defects of the Hand1/Hand2 mutants, can the authors expand their speculation in the discussion over genetic pathways likely abrogated by the Hand1- Hand2 switch that result in the cardiac non-compaction phenotype?*

We now expand discussion of the published *Hand1* and *Hand2* loss of function cardiac studies and based on these phenotypes conclude that the *Hand1*^{Hand2/Hand2} phenotypes could include some gain-of-function effects but given we have identified cardiac gene expression regulated in *Hand1* loss-of-function models that is not restored in *Hand1*^{Hand2/Hand2} hearts, we feel this supports the idea that HAND2 is not fully rescuing loss of myocardial HAND1, which we speculate further is altering myocardial sourced signaling that is associated with *Hand* gene expression (BMP/TGF, WNT, and SHH) could be resulting in non-compacted *Hand1*^{Hand2/Hand2} mutant hearts. We would rather not speculate further given we did not examine cardiac gene expression and plan to do this in future studies of this mouse model. We hope this satisfies the Reviewers concern.

Reviewer 2:

The authors present significant and novel findings on these two important developmental transcription factors, HAND1 and HAND2. First, revealing important findings on both in vivo functional redundancies and also uncovering novel functions in intestinal development.

We thank Reviewer 1 for these highly supportive and encouraging comments.

SUGGESTIONS TO AUTHORS

The manuscript by Firulli et al investigates the functional redundancy of the HAND1 and HAND2 transcription factors. Specifically, mice in which the Hand2 exons/intron were inserted

into the *Hand1* locus (replacing *Hand1* exons/intron) were generated. Phenotypic and transcriptomic analysis demonstrated that *Hand2* was able to rescue the extraembryonic defects found with complete deletion of *Hand1* supporting redundancy in this aspect of *Hand1/2* functions. Interestingly, the subsequent analysis of these mice (*Hand1Hand2/Hand2*) demonstrated early lethality and an intestinal developmental abnormality, omphalocele, supporting a unique role for *Hand1* in this developmental process. Molecular analysis using RNA in situ hybridization and spatial transcriptomics identified dysregulation of *Wnt* signaling and *Pitx2* as potential mediators downstream of *Hand1*. Additional analysis of the hearts of E16.5 *Hand1Hand2/Hand2* embryos demonstrated cardiac defects including thin myocardium and ventricular septal defects and mild cardiac conduction defects in adult survivors consistent with the inability of *HAND2* to fully replace *HAND1* function in later stages of heart development and function. This was further supported by the identification of mild conduction abnormalities in mice harboring heterozygous for LV enhancer deletion of *Hand1* and the *Hand1Hand2* alleles (*Hand1 Δ LV/Hand2*). I have some minor comments which are detailed below.

1. Results section in manuscript and Table 1. Clarify results of intercrossing between *Hand1*^{+/+}/*Hand2* mice. Unclear why only 1-3 pups were found per litter as 25% should be expected to be *Hand1Hand2/Hand2*? Unclear if this is due to lethality of *Hand1*^{+/+}/*Hand2* mice or was the small litter size due to excessive maternal cannibalism?

We apologize for the confusion. The 1-3 pups denoted in the manuscript refers to *Hand1*^{*Hand2/Hand2*} intercrosses not the breeding of the heterozygotes that populate the data on table 1. While re-reading the sentence, I can see where the confusion comes from and have now edited it to improve clarity.

2. Figure 3. Clarify the total number of embryos examined for gut rotation in the Figure legend and from how many litters these animals were collected.

Thank you for pointing this out. We now add this information to the Figure 3 legend defining that we examined 12 control, 14 heterozygous, and 14 *Hand1*^{*Hand2/Hand2*} homozygous embryos from 5 litters.

3. Figure 4, re-organize the Control and *Hand1Hand2/Hand2* images consistent for A-H and I-P so Control is always on left.

Thank you, for the comment. Our reasoning for setting up Figure 4 in this was to place the mutant sections next to each other for easier comparisons. We now update Figure 4 to have the data ordered: Control, Mutant, Control, Mutant in 4 columns with more appropriate A-P labels. We hope this addresses the clarity and consistency concerns.

4. Figure 4, include the N for these experiments in the figure legend

Each probe was assayed on 3-5 embryos; this is now added to the Figure 4 Legend.

6. Legend Figure 6A. Correct typo where "patent" interventricular septum is mentioned as septation is completed in control *Hand1*^{+/+} hearts.

We fixed the sentence to read a fully developed interventricular septum as suggested.

7. Figure 6G. A select number of cardiac genes are examined at E11.5 by qRT-PCR, the addition of rationale for choosing these genes in manuscript text would be helpful for the reader

Thank you for the comment. We chose these genes based on what we observed as being regulated in the *Hand1* cardiomyocyte conditional knockout (Firulli et al 2019, Cardiovascular Research). We now state this in the results paragraph referring to Figure 6G.

8. Table 3. Clarify the number of animals examined as in the Table n=50 but in the manuscript

text n=57.

Apologies for the typo. We have adjusted the table to the correct number (57). Thank you for catching this error.

9. *Fig. 7 and Supp Fig. 7. EKG abnormalities could be better represented to improve clarity. Some suggestions include: 1. Heart rate is described in manuscript text while associated Figure 7A shows on RR intervals, consider using one value to represent heart rate 2. Significance of QRS abnormalities that are specific to leads could be clarified, are they representative of right vs left bundle branch block and how do these findings compare to the phenotype of mice with loss of Hand1 in the conduction system.*

Apologies, we have edited the results and discussion sections to comply with this request.

10. *Scale bars should be added for Supp. Figures 1B-1G, 2A-C,*

We now add a scale bar that defines the scale for all the images in Supplemental Figure 1B-G as well as scale bars for each image in Supplemental Figure 2.

11. *Results, Section titled "Transcriptomic analysis comparing...". Define "Hand1fx" in text or Figure legend*

We now define and cite the source of the *Hand1* conditional knockout mice in this section.

12. *Correct typo in Figure Legends. "Hand2Hand2/Hand2" should be "Hand1Hand2/Hand2"*

Thank you for catching these embarrassing typos. These are now corrected.

13. *Correct typo in manuscript, Results, section titled "Hand1Hand2/Hand2 mice are viable...", para 2, 2nd to last sentence. (ens; Figure 4A, A') should be Fig S5A, A')*

Thank you, we have fixed this typo.

Reviewer 3:

In their submitted manuscript, Firulli and colleagues present a Hand2 ORF substitution in the mouse Hand1 locus to address long-standing questions of redundancy between these two core transcription factors involved in cardiac and other development. The authors report several phenotype readouts under different genetic conditions. The data is complemented by spatial transcriptomics with main focus on omphalocele phenotypes connected to defects with ventral body wall closure.

Overall, the manuscript adds to clarifying key unknowns in the field concerning the redundancy vs unique functions of Hand1 vs Hand2 during mammalian development. Nonetheless, in its current write-up, it unfortunately falls short of its intended purpose. More so, the manuscript is hard to follow as it jumps between different experiments, timepoints, allelic combinations, and tissue focus. For instance, the first paragraph of the intro has hardly anything to do with the overall scope of the work. Making better use of the intro space and discussion would already greatly improve the manuscript.

We thank Reviewer 3 for the supportive and encouraging comments noting that we add *clarifying key unknowns* and hope that our revision of the manuscript writing based on the comments from all of the Reviewer's improves enthusiasm for this work. We have, as suggested, edited the introduction to discuss the published supporting data on why HAND factors are considered functionally redundant. We also have edited the Discussion based on comments from all three Reviewers to further address this concern.

Big picture-wise, the authors state that their experiments show Hand1 and Hand2 to be broadly redundant - which is a claim that is not properly backed up by the presented data, as

well as outlined with contradictory statements throughout the manuscript. For instance, the authors obtain only 2% survival with their substitution in a Hand1 null background: while the authors then go on to show this severely diminished outcome is due to intrauterine omphalocele and embryo loss, the context for their interpretation of redundancy is (as it seems) heavily influenced by the authors' deep knowledge of the topic and less by a rational chain of events outlined by the presented data. The mechanistic data on cardiac conductivity and function seem disconnected and are once more not placed in context to explain why these phenotypes are followed up.

We would like to remind Reviewer 3 that *Hand1* systemic loss-of-function is embryonic lethal at E9.5 and the gene replacement employed in this study does in fact rescue the main phenotype of these mice, the extraembryonic defects, to birth and 2% of homozygous mutant pups are born without omphalocele allowing adult survival. The pups with omphalocele are born live but are eaten by the mothers cleaning up the newborn pups. Additionally, the bulk yolk sac RNA-seq data shows robust restoration of gene expression altered by loss of *Hand1* when *Hand2* replaces *Hand1* in the yolk sac and this rescue with the expression of *Hand2* is accompanied by a yolk sac phenotype rescue. The finding that 2% of *Hand1*^{*Hand2/Hand2*} neonates born without omphalocele survive into adulthood, further supports (backs up) the conclusion that there is broad functional redundancy. We ask Reviewer 3 to take into consideration that Reviewers 1 and 2 do not share this criticism of our conclusions and combined with the direct observation that there is a significant restoration of embryo survival from E9.5 to live birth, that Reviewer 3 concedes that the presented data in this manuscript indeed does “properly back up” the conclusion that HAND2 holds partially redundant functions with HAND1.

We apologize that we did not make the link to the importance of cardiac conduction in surviving mutants clearer. *Hand1* loss-of function within the developing left ventricle (*Hand1*^{*ΔLV/ΔLV*}; Vincentz et al. 2019) results in significant morphological and functional cardiac conduction defects. Based on these significant published findings, interrogating conduction function within the *Hand1*^{*Hand2/Hand2*} and *Hand1*^{*Hand2/ΔLV*} mice is a logical and important investigation for truly examining HAND functional redundancy in the heart. We have edited the results section to make this point clearer and hope that now this data better connects within the revised manuscript.

Importantly, the paper write-up is highly specialized in its outline of used reagents, genetic backgrounds, readouts, as well as in how the resulting data is interpreted. With minimal to no introduction to previously established phenotypes associated with Hand1/Hand2 mutants, the manuscript is only accessible to deep experts. The figures are minimalistic at times and several critical data points are shoved into the supplemental data, where they do not belong (e.g., Supplementary Fig 1 is main figure material). Furthermore, several claims throughout the manuscript are interpretative and not presented with quantifications or comparisons to underline the significance. Many issues of the manuscript can potentially be salvaged, however, with a re-write and revisited presentation.

We refer to comments above addressing other Reviewers on our expansion of the information regarding the reagents, genetic backgrounds, readouts in the results, and discussion. We appreciate that Reviewer 3 feels Supplemental Figure 1 should be included as a main Figure. I hope Reviewer 3 can appreciate that *Development* allows for only 7000 words in the main body text including tables and figure legends. With 8 main Figures, 3 Tables, 8 Supplemental Figures and 8 Supplemental Files, it is not possible to include every data Figure within the main body of the manuscript. Given that Supplemental Figure 1 depicts the construct design and test of *Hand* gene expression within the *Hand1*^{*Hand2/Hand2*} mouse model and in the revision, test of the potential hybrid mRNA that Reviewer 1 astutely pointed out, we hope Reviewer 3 can understand why this Figure is better suited for supplemental data. We have, based on the comments of all 3 Reviewers, edited the manuscript accordingly and hope it is considered much improved.

Additional Major Points:

1) *Genetics: For their Hand1/2 substitution, the authors claim this is not an over-expression - but with an additional copy of Hand2 that is exactly what this is. The authors*

even explicitly state that there is a two-fold increase in *Hand2* expression in the *Hand1Hand2/Hand2* mice. This should be re-phrased and re-interpreted overall. This connects also to the overall statement as to how can these genes be functionally redundant when there's a 2% survival rate when you get rid of one of them?

We would like to clarify this comment regarding *Hand2* expression within *Hand1* expressing tissues. In the *Hand1^{Hand2/Hand2}* mice there is no *Hand1* expression (Supplemental Figure 1). *Hand2* expression replaces the missing *Hand1* expression and the replacement *Hand2* is transcribed by the *Hand1* cis-regulatory elements. In tissues such as the extraembryonic mesoderm, where *Hand2* is not expressed, we can observe that the resultant extraembryonic phenotype and the yolk sac gene expression is largely restored by this replacement.

In Supplemental Figure 1L E9.5 whole embryos were evaluated for *Hand1* and *Hand2* expression by qRT-PCR. As *Hand1* message lowers *Hand2* message is increased. By looking at wild type controls if you consider total *Hand* message as 2-fold (1-fold *Hand1* 1-fold *Hand2*) when looking at *Hand1^{Hand2/Hand2}* embryo data, the net *Hand* expression is very close to 2-fold (0-fold *Hand1* and 2-fold *Hand2*). We acknowledge that there is slightly more *Hand2* in our heterozygous *Hand1^{Hand2/+}* embryo data a reason we did not use this allelic combination in this study beyond what we show in Sup Fig. 1L. We further point out that in tissues where *Hand1* and *Hand2* are co-expressed such as lateral mesoderm and some myocardium, that there is more *Hand2* message in these tissues, but these tissues have no *Hand1* expression so the net *Hand* expression within these tissues is still similar to controls.

An additional piece of supporting data which leads to the conclusion that there is not a significant overexpression of *Hand2* is the E13.5 Visium data DE analysis (Supplemental File 8). Examining *Hand2* expression within each of the 8 clusters, we observe only clusters 2 and 5 show a significant increase in *Hand2* expression (again accompanied by a significant decrease of *Hand1*), which is less than 1 Log2FC (so under 2-fold). *Hand2* levels in other clusters are not significantly affected. We now discuss this more extensively in the results section pertaining to Figure 5. We now hope this more detailed evaluation of the obtained *Hand* expression within our data now makes our conclusions clearer and better supported within the revised manuscript.

2) *Biological basics and background knowledge: Connecting to the challenging accessibility of the paper as presented, the figure showing the expression patterns of Hand1 and Hand2 in their respective mutants should be in the main paper and not a supplement. Furthermore, a proper outline of all used alleles and their phenotypes as originally reported, e.g., in the introduction (Hand1LacZ, etc.), would greatly strengthen the manuscript.*

Again, we greatly appreciate that Reviewer 3 would like to see more of the supplemental data within the main body of the text. Given that *Development* provides guidelines of manuscript length and number of allowed Figures, we feel our choices for what Figures should be within the manuscript and which are more suitable for supplemental data is appropriate as the data specified in this comment are simply a confirmatory test of the efficacy of the new mouse allele. Given that Reviewers 1 and 2 do not share this concern, the Authors hope it is acceptable to keep the organization of the paper data as is. We do fully agree that a better description of the mouse models is a great idea, and we now include a supplemental table (Supplemental Table 1) with the various mouse alleles employed in this work along with their citations.

3) *Phenotype interpretation: The authors claim the phenotype in the Hand1LacZ/LacZ is due to a yolk sac deficiency; however, they provide no evidence that this hypoplasia is solely due to the yolk sac deficiencies as opposed to any other developmental tissue.*

The *Hand1* systemic knockout was published in 1998 by me (ABF) when I was a postdoc, and independently by Paul Riley when he was a postdoc in Jay Cross's lab. The evaluation of the *Hand1* systemic loss-of-function phenotype and cause of death reported in both papers is extraembryonic defects within the visceral mesoderm of the yolk sac is well documented in these two peer-reviewed published *Nature Genetics* manuscripts that are cited in this study. Based on comments regarding the poor focus of the introduction, we now define these

published findings more clearly.

4) *Connection of the write-up with the data: throughout the paper, the text seems to be a train-of-thought text without incorporating all links to the actually obtained data. Case in point, the entire second paragraph of the section entitled "Transcriptomic analysis comparing Hand1Hand2/Hand2 and H1CKO yolk sacs reveals rescue of the extraembryonic Hand1 loss-of function phenotype.", the authors reference supp data but never once call out a figure or have that data represented in a main figure.*

We again refer to the word limits of a *Development* manuscript. We apologize that we cannot add additional Figures to the manuscripts main body without exceeding these limits or eliminating what the Authors see as the critical data to support the conclusions. To be clear, the first paragraph of the noted section by Reviewer 3 in this comment directly lays out the data found in primary Figure 1. The second paragraph that Reviewer 3 refer to describes the IPA analysis of the data found in Figure 1. These IPA analyses, spreadsheets compare *H1CKOs* and *Wild type* (Supplemental File 2), *Hand1^{Hand2/Hand2}* and *Wild type* (Supplemental File 3) and *H1CKOs* and *Hand1^{Hand2/Hand2}* (Supplemental File 4) and are discussed within the results section 2nd paragraph. It is unfortunately difficult to be responsive to this comment as the Authors do not see how three 11-page IPA evaluations could be constructed into an easily interpretable primary Figure that includes all the data from each analysis. We kindly ask Reviewer 3 that as Review 1 and Reviewer 2 do not criticize our use of Supplemental data to help support this work that we be allowed to maintain our current data format with responses to writing and data presentation updated as directed.

5) *How meaningful are the interpretations of the intestinal curvature when dissected out of E17.5 mice? How faithful is the organ morphology when removed from the animal? Additionally, there seems to be an inconsistency among the control mice as well, such that interpreting differences in intestine looping feels dubious. In addition, the authors state this data matches with the observed high frequency of omphalocele in their mutants, but only observed a subtle curvature phenotype in 2 of 4 mice. Clarifying and contextualizing this data would greatly strengthen the work.*

Based on the literature that is cited within this manuscript the Authors are confident that the looping of the gut is a well regulated consistent biological morphological process, and the interpretations are valid especially considering the data in Figure 2. Removing the stomach and intestines was employed biased on consultation with leading expert in gut rotation (Natasza Kurpios see acknowledgements) as the proper way to evaluate gut rotation. It is unclear what inconsistencies Reviewer 3 sees that Review 1 or 2 did not? The literature clearly defines that malrotation of the developing gut is an established cause of omphalocele. Note, we observe omphalocele in the majority *Hand1^{Hand2/Hand2}* neonates, and we also observe changes in L/R patterning genes (Fig. 4) that are established to influence gut rotation. The citations from our manuscript that demonstrate this critical important work are listed below for convenience.

Sanketi, B. D., Zuela-Sopilniak, N., Bundschuh, E., Gopal, S., Hu, S., Long, J., Lammerding, J., Hopyan, S. and Kurpios, N. A. (2022). Pitx2 patterns an accelerator-brake mechanical feedback through latent TGFbeta to rotate the gut. *Science* 377, eabl3921.

Welsh, I. C., Thomsen, M., Gludish, D. W., Alfonso-Parra, C., Bai, Y., Martin, J. F. and Kurpios, N. A. (2013). Integration of left-right Pitx2 transcription and Wnt signaling drives asymmetric gut morphogenesis via Daam2. *Developmental cell* 26, 629-644.

6) *Related, the authors have previously contributed to establishing mesothelial expression of Hand1/2 - this tissue associates closely with ventral body wall closure (e.g., work from the Schultheiss lab in chick and mice), which the authors however seem to skip in favor of following anomalies with inner organs. Providing more context to the omphalocele phenotype would greatly expand the scope of the work and involvement of Hand1/2.*

Reviewer 3 is correct that we contributed to a recent study on mesothelial expression of *Hand2* in the zebrafish, which only contains a single *Hand* gene more closely related to *Hand2*. We also agree that body wall closure is also a possible mechanism for omphalocele in addition to the gut rotation mechanisms that we have direct evidence for. We also note that body wall

closure and gut rotation are not mutually exclusive mechanisms. To fully interrogate mesothelial mechanisms that could be contributing to the observed phenotype would require significant time to research as well as it would add more data to a manuscript that this Reviewer contends holds too much data in supplements. Our inclusion of *Tbx20* as a gene feature plot in Figure 5 is due to its role in mesothelial mechanism body wall closure that is a focus of the Kahn lab (who we thanked in the acknowledgements). Based on comments tendered below, we looked carefully at the DE analysis from the Visium (Supplemental file 8) and find that *Tbx20* is not significantly regulated within mesothelium or any cells within the data. In personal communications with the Kahn lab that I will share, we sent them our *Hand1* conditional allele so they could employ their body wall-restricted *Cre* driver that expresses *Cre* in the ventral body wall to delete *Hand1*. Preliminary data show that the phenotype is not omphalocele. Given this personal knowledge that I share in confidence, we feel it would be inappropriate to invest additional time on exploring mesothelial mechanism in this manuscript but do feel continuing to explore mesothelial involvement is an important future endeavor. We now add in our discussion that body wall closure could contribute to the omphalocele phenotype and cite the manuscript we seemed to skip.

7) *The gene expression panel following the spatial transcriptomics is presented without quantification of sample size or expression levels, but merely provided as overall interpretations. Quantification of expression levels (also compared to the Visium data) would strengthen the interpretation of these datapoints as well.*

We apologize for some confusion with this comment. The data following the spatial experiment (Figure 5) is evaluation of the adult heart phenotypes encountered and qRTPCR analysis of known *Hand1* regulated cardiac targets (Figure 6). Sample size for the Visium analysis is defined within the methods: “four E13.5 *Hand1*^{Hand2/Hand2} and four *Hand1*^{+/+} E13.5 embryos (4-sections per animal on each Visium slide). Sample size of Figure 6 data is also defined in its legend and cannot be compared to the Visium data as it is different tissue and timepoints. If Reviewer 3 is referring to the “previous expression panel” i.e. Figure 4 showing RNAscope data at E10.5, this makes more sense. If this is the case, we have based on comments above, edited the legend and the Figure 4 to improve clarity adding by the number of samples evaluated in this semiquantitative assay. Given Figure 4 analysis is at E10.5 and the Visium at E13.5, we employed caution in direct comparison within the revised text.

To try to get this right, we also have edited the Figure 5 legend text to reinforce the sample information. Note the Visium data is fully included in supplemental files 5 and 6 with images of the Visium slide sections. We now also add the log2FC and adjusted p value within the text of the discussion for the DE analysis. We thank Reviewer 3 and apologies again for our confusion with this comment.

8) *The cardiac function assays require better contextualization and connection with what has previously been shown about Hand function (which the authors know expertly well, but not every reader).*

We refer above to our comments addressed above where edits making *better* contextualization and connection of the published *Hand1* allele phenotypes are now included.

Minor points:

a) *Developmental stages of performed experiments are widely variable, with several experiments being compared at different timepoints without explanation as to what the rationale is or how these time points fit together overall. Rationalizing the time points would greatly help in selling the manuscript to the reader.*

This is an appreciated comment. In all eight primary Figures, we are directly comparing the same stage embryos to each other. We do not feel we are directly comparing experiment to experiment; however, we do discuss the gene expression of some genes at more than one time point (example *Pitx2* at E10.5 in Figure 4 and E13.5 in Figure 5) within that Figures data set. Stage choice in this study is important and was chosen for biological relevance based on the employed assay. We now add a detailed rational for the chosen time points within the legend of the new Supplemental Table 1. Copied below for convenience and thank Reviewer 3 for their

input

Supplementary Table I: Description of the mouse alleles employed in this study. Animal stage choice rational employed in this study are as follows: E9.5 is chosen for yolk sac expression data (Fig. 1) as this is when *Hand1*^{LacZ/LacZ} embryos begin to die. E16.5 is chosen for H&E section analysis for omphalocele (Fig. 2) as this is stage when body wall closure is completed. E17.5 (Fig. 3) was chosen to look at dissected stomach and intestine gut rotation as the process is completed by this stage. Left-Right gene expression is observable at E10.5 within the dm and is an ideal time point for scoring early altered malrotation gene expression (Fig. 4). E13.5 is several days prior to body wall closure thus optimal time point for evaluating gene expression relevant to omphalocele (Fig 5). E16.5 hearts were evaluated as intraventricular septum closure is fully completed by this stage (Fig. 6). Adult animals were employed for adult mouse conduction evaluation (Fig. 7).

b) *The in-text description of Fig. 5B does not appear to be a complete sentence.*

We have fixed the sentence.

c) *The method of gene expression evaluation is not mentioned (qPCR?). The authors are further encouraged to have a more specific label than "relative quantification" in 6G.*

We indeed used QRT-PCR to evaluate the gene expression of the candidate cardiac gene in Figure 6G and is stated in the Figure 6G legend of the original and resubmitted version of the manuscript. Our qPCR cyclers a QuantStudio 3 (Applied Biosystems) quantitative thermocycler software calculates relative quantitation in addition to employing Benjamini-Hochberg False discovery rate to determine significance. Error bars denote the maximum and minimum relative level of gene expression in the test samples calculated using the confidence level set in the QuantStudio 3&5 software analysis settings. n≥6 in all experiments for all genotypes assayed. This is detailed within the methods section of the original and resubmitted manuscript.

d) *Figure 1 A, right label is overlapping with volcano plot (in downloaded high res version).*

We apologize for the placement of the key and have moved it to the top of the panel.

e) *A few key typos of alleles, as well as inconsistencies in spelling of genetic features (lacZ vs LacZ, etc.).*

We have made an honest effort to fix the typos.

We greatly appreciate the time and insights that all the Reviewers provided to us in their evaluation of our manuscript. We hope that our edits and additions to what was considered an elegant, significant, and clarifying study is now suitable for publication in *Development*.

Second decision letter

MS ID#: dev.204963R1

MS TITLE: Hand1 gene replacement with Hand2 reveals overlap in function with unique occurrence of omphalocele and heart defects.

AUTHORS: Beth A Firulli; Chloe A Ferguson; Corrie de Gier-de Vries; Ram Podicheti; Douglas B Rusch; Vincent M Christoffels; Michael Rubart-von der Lohe; Anthony B Firulli

Dear Dr Firulli,

Based on your response to reviewer, we would like to publish a revised manuscript in Development. I find you have addressed the comments adequately throughout. I encourage you to include supplementary figure 1 as a new main Fig 1, as the casual reader would likely be interested to see the strategy and validation. Also, in supplementary figure 2 please change "principle" to "principal". For the aficionados, I also encourage you to include the ECG tracings from Suppl Fig 7 in the main figure 7. In Fig 6, Anf should be Nppa. Also consider in current Fig 1 to add more rescued transcripts in bar graph format, which would take up less space while increasing information for the reader.

Second revision

Author response to reviewers' comments

Based on your response to reviewer, we would like to publish a revised manuscript in Development. I find you have addressed the comments adequately throughout.

We are extremely grateful for the recognition of our efforts to respond to the Reviewer comments.

I encourage you to include supplementary figure 1 as a new main Fig 1, as the casual reader would likely be interested to see the strategy and validation.

We have incorporated Figure S1 as Figure 1 in the revised Manuscript.

Also, in supplementary figure 2 please change "principle" to "principal".

Apologies this typo is now fixed.

For the aficionados, I also encourage you to include the ECG tracings from Suppl Fig 7 in the main figure 7.

Thank you for the suggestion. We have added the ECGs to Figure 7.

In Fig 6, Anf should be Nppa.

Apologies, we have changed the label of Anf to Nppa and adress this in the text.

Also consider in current Fig 1 to add more rescued transcripts in bar graph format, which would take up less space while increasing information for the reader.

We like this suggestion but had a hard time creating a bar graph that would show the sample points within the bar without looking too busy. What I came up with is a single graph that now includes 10 genes showing each individual yolk sac sample (similar to the original). I employed `ggplot` in R-studio and think it meets the suggestion of taking up less space and simpler to read. We now color code each genotype which improves evaluation of the data. We also edited the results section to reflect the updated data and Figure 2.

We thank the Editor for their time and critical eye for what is now a clearer and more focused story.

Third decision letter

MS ID#: dev.204963R2

MS TITLE: Hand1 gene replacement with Hand2 reveals overlap in function with unique occurrence of omphalocele and heart defects.

AUTHORS: Beth A Firulli; Chloe A Ferguson; Corrie de Gier-de Vries; Ram Podicheti; Douglas B Rusch; Vincent M Christoffels; Michael Rubart-von der Lohe; Anthony B Firulli
ARTICLE TYPE: Research Article

Dear Dr Firulli,

I am happy to tell you that your manuscript has been accepted for publication in Development, pending our standard publication integrity checks.